# Are Engineered Geothermal Energy Systems a Viable Solution for Arctic Off-Grid Communities? A Techno-Economic Study

**Mafalda M. Miranda [1,\*], Jasmin Raymond [1], Jonathan Willis-Richards [2] and Chrystel Dezayes [3]**

1 INRS—Institut National de la Recherche Scientifique, 490 Rue de la Couronne, Québec, QC G1K 9A9, Canada; jasmin.raymond@inrs.ca
2 Fracture and Reliability Research Institute, Tohoku University, 2 Chome-1-1 Katahira, Aoba Ward, Sendai 980-8577, Japan; ej.willisrichards@gmail.com
3 BRGM—Bureau de Recherches Géologiques et Minières, 45060 Orléans, France; c.dezayes@brgm.fr
\* Correspondence: mafalda_alexandra.miranda@inrs.ca

**Abstract:** Deep geothermal energy sources harvested by circulating fluids in engineered geothermal energy systems can be a solution for diesel-based northern Canadian communities. However, poor knowledge of relevant geology and thermo-hydro-mechanical data introduces significant uncertainty in numerical simulations. Here, a first-order assessment was undertaken following a "what-if" approach to help design an engineered geothermal energy system for each of the uncertain scenarios. Each possibility meets the thermal energy needs of the community, keeping the water losses, the reservoir flow impedance and the thermal drawdown within predefined targets. Additionally, the levelized cost of energy was evaluated using the Monte Carlo method to deal with the uncertainty of the inputs and assess their influence on the output response. Hydraulically stimulated geothermal reservoirs of potential commercial interest were simulated in this work. In fact, the probability of providing heating energy at a lower cost than the business-as-usual scenario with oil furnaces ranges between 8 and 92%. Although the results of this work are speculative and subject to uncertainty, geothermal energy seems a potentially viable alternative solution to help in the energy transition of remote northern communities.

**Keywords:** FRACSIM3D; shear displacement–dilation model; poroelasticity; levelized cost of energy; Monte Carlo method; geothermal energy; subarctic; Nunavik

## 1. Introduction

The lack of an environmentally benign energy supply for electricity, space heating, domestic hot water and cooking is still a reality, not only in the off-grid communities of Canada, but worldwide [1]. Fossil fuels have been the main sources of electricity, space heating and cooking fuels in the developed world. However, current climate change concerns, people's well-being and the environment are changing this predominance. Clean energy supply and sustainability are buzzwords nowadays. The worldwide consumption of renewable energies has increased from 40 EJ in 1990 to 64 EJ in 2017 [1]. Although smaller than the consumption of non-renewables (304 EJ in 2017; [1]), this is an important increase. Off-grid renewable energy solutions, including stand-alone systems and microgrids, are often a viable electrification solution [2].

In Canada, for example, from the approximately 280 off-grid communities, 239 rely exclusively on diesel for electricity, space heating and domestic hot water [3]. Within the diesel-based settlements without road access, the fuel must be imported during summer and stored for year-round use [4]. Such an energy situation entails significant costs, low energy security and a high probability of damaging an already vulnerable ecosystem. Therefore, interest in assessing the potential for renewable sources to feed microgrids in remote communities has increased and several studies have been conducted (e.g., [5–17]). Among these options, deep geothermal energy sources can play a key role to provide

baseload power and/or heat to the off-grid settlements (e.g., [18–27]). In fact, a first-order community-scale geothermal assessment undertaken in Kuujjuaq (Nunavik, Canada) suggested that the deep geothermal energy source can fulfil the community's annual average heating demand of 37 GWh [28]. This community is settled on the Canadian Shield, a physiographic region that has been considered a target for geothermal exploration through engineered/enhanced geothermal systems [25,29]. The feasibility of such systems has been studied in the Western Canadian Sedimentary Basin and Arctic Lands [18,19,25,30–32], but few studies have been conducted in the Canadian Shield [24,25], where there are hundreds of off-grid communities relying heavily on fossil fuels [4,21]. Therefore, there is a need for a comprehensive estimation of the possible performance of deep engineered geothermal energy systems across the Canadian Shield. In this context, work has been carried out in the community of Kuujjuaq (Nunavik, Canada) using the limited local surficial and regional data available to provide first-order answers to the following key questions:

1.  Will the hydraulic stimulation technique applied in crystalline basement rocks elsewhere develop a well-connected flowing system in Kuujjuaq? How can this be done? What further local geological and thermo-hydro-mechanical data is required for more accurate predictions?
2.  Are the deep geothermal energy sources harvested by engineered geothermal energy systems in Kuujjuaq likely to be cost-competitive compared to fossil fuels?

The answers to these questions are, however, subject to high uncertainties due to the current poor knowledge of both geology and thermo-hydro-mechanical data. Unfortunately, no hydraulic stimulation field experiments have been carried out to date in Kuujjuaq, nor were they within the scope of the present study. Thus, no history matching is available to calibrate the numerical simulations carried out in this work. Nevertheless, a "what-if" approach was used to provide a range of possibilities and help to design an engineered geothermal energy system for each of the uncertain scenarios. Each possibility aims to provide the thermal energy needed for the community, while keeping the circulating water losses below 20%, the reservoir flow impedance below 1.0 MPa $L^{-1}$ $s^{-1}$ and the system thermal drawdown below 1 °C/year. Thus, this study offers a first-order prediction of the performance of engineered geothermal energy systems as off-grid solutions (and constraints on the geomechanical and geological assumptions required) to help in the energy transition of remote northern communities. Additionally, a preliminary evaluation of the levelized cost of energy was undertaken to forecast the economic potential of engineered geothermal energy systems in remote northern regions. Overall, this study may help trigger interest for further geothermal exploration which is fundamental for an accurate evaluation of the deep geothermal energy potential beneath off-grid settlements.

Thus, this study was motivated by the lack of clean energy supply in the majority of the Canadian remote northern communities. The goal was to assess if deep geothermal energy harvested by engineered geothermal energy systems, or enhanced geothermal systems, is a technically and economically viable alternative solution to offset the diesel consumption in such communities. The study was undertaken in the off-grid settlement of Kuujjuaq (Nunavik, Canada) to provide an example for the remaining off-grid communities. The study here described, and the results obtained, although highly speculative due to the lack of deep geothermal exploratory boreholes in the study area, represent an important contribution to understand the potential that deep geothermal energy sources have to offer in the energy transition of diesel-based regions of northern Canada and other arctic regions. Furthermore, this study aims to predict the performance of enhanced geothermal systems in a location with significant geothermal data gaps. This may raise awareness about the potential of geothermal energy in areas considered at first sight unviable and trigger the interest for further geothermal developments. The parameters that require further data collection and possible operational strategies to develop engineered geothermal energy systems for communities on the old Canadian Shield are highlighted in this manuscript.

## 2. Background Information

### 2.1. Engineered/Enhanced Geothermal Systems

Enhancing well productivity and the permeability of subsurface rocks has been common practice in the oil and gas industry for many years (e.g., [33]). However, it was only introduced in geothermal energy exploration in 1973. The first hydraulic stimulation experiments in crystalline rocks were done at Fenton Hill (e.g., [34]). The success of these field experiments opened new opportunities to explore geothermal energy sources in areas that were previously considered unviable (e.g., crystalline rocks with very low permeability). In fact, this new concept of recovering the Earth's heat via a pressurized closed-loop circulation of fluid from the surface through a hydraulically stimulated and confined reservoir several kilometers deep made in crystalline basement rocks marked an important conceptual turning point in the geothermal energy industry.

Since the 1970s, several projects have been started worldwide, applying different stimulation techniques and in geological contexts ranging from crystalline to sedimentary rocks. A review of these projects is given by, for instance, Tester et al. [35], Breede et al. [36], Xie et al. [37], Olasolo et al. [38], Lu [39] and Kumari and Ranjith [40]. This research increased knowledge of the behavior of rock masses and joints subjected to hydraulic stimulation. The current successful commercial projects (e.g., Soultz, Landau and Rittershoffen) were built upon this previously gained know-how. Furthermore, field-scale underground laboratories (e.g., Grimsel, EGS Collab and Utah FORGE) are tackling hydro-thermal-mechanical questions that have remained unresolved in the past. Moreover, although only a few sites are generating geothermal energy power (e.g., Soultz, Landau and Rittershoffen), all of the abandoned, suspended and ongoing projects are still important research facilities, providing a significant scientific database. For example, the Fenton Hill venture is described in great detail by Brown et al. [34] and a summary of the lessons learned can be read in Kelkar et al. [41]. A compilation of the development phases of the Rosemanowes geothermal project, problems faced and unresolved issues are provided by, for example, Parker [42]. Richards et al. [43] discuss the performance and characteristics of the Rosemanowes hydraulically stimulated geothermal reservoir. The authors also discuss the fundamental parameters controlling the impedance, thermal performance and water losses. The contribution of the Soultz project for the scientific community and its development phases are described in detail by, for instance, Genter et al. [44]. The lessons learned from past geothermal projects employing hydraulic stimulation treatments to crystalline rocks were taken into consideration in this study.

### 2.2. Nunavik's Geothermal Potential

The geothermal energy potential of Nunavik and northern Québec has been investigated by Majorowicz and Minea [24] and Comeau et al. [45]. These regional-scale studies are based on scarce and sparse data distribution (Figure 1). In fact, only three deep boreholes with heat flow assessment exist in Nunavik (Camp Coulon, Raglan and Asbestos Hill mining sites), another one is in Nunavut (Nielsen Island) and one in Newfoundland and Labrador (Voisey Bay). All of these boreholes lie at distances of 430 to 500 km from Kuujjuaq. The heat flow estimated in these five sites ranges from 22 to 38 mW m$^{-2}$ (Figure 1). Moreover, Comeau et al. [45] inferred the 1D subsurface temperature distribution among the different geologic provinces of Québec and their results suggest that at 5 km depth, the temperature in the Churchill Province ranges from 49 to 53 °C. However, only two data points exist for the Churchill Province, and these lie at a distance of almost 500 km from Kuujjuaq (Figure 1). Extrapolating such values to Kuujjuaq appears unrealistic, justifying a community-based approach to evaluate the local geothermal potential.

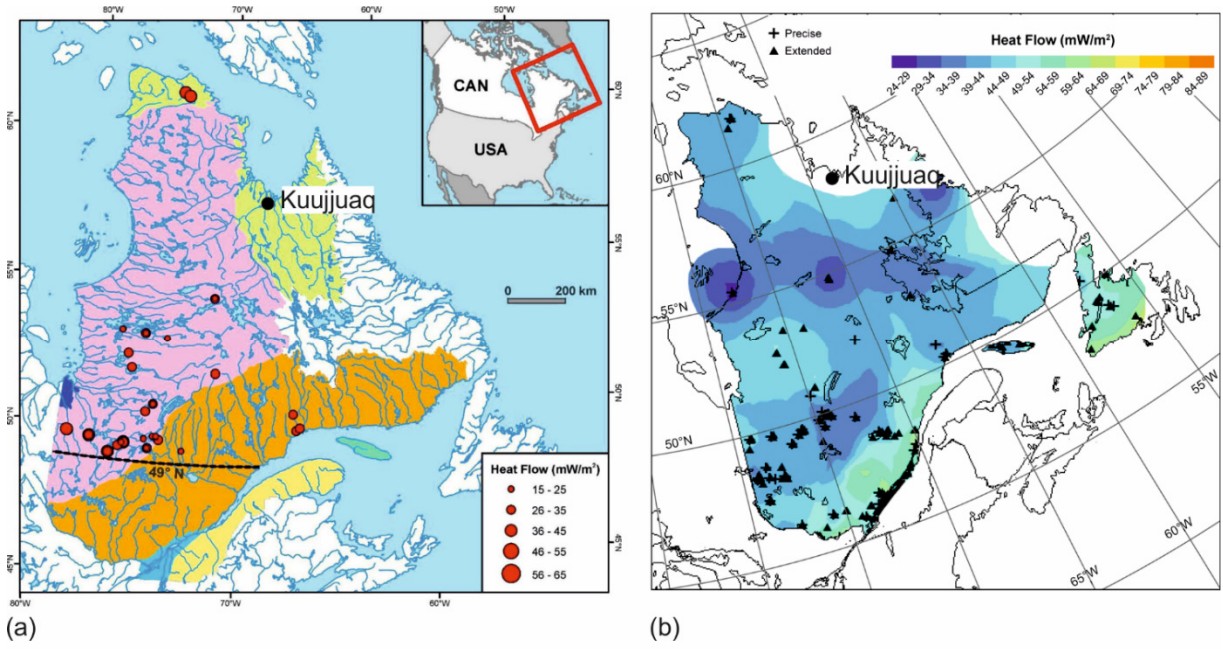

**Figure 1.** (**a**) Distribution of heat flow data available and (**b**) geothermal heat flow density for northern Québec. In (**a**) pink—Superior Province, green—Churchill Province, orange—Grenville Province, yellow—Appalachian Province, dark blue—Hudson Bay Platform and light blue—St. Lawrence Lowlands. Redrawn from Majorowicz and Minea [24] and Comeau et al. [45].

## 3. Geographical and Geological Setting

The settlement of Kuujjuaq, located north of the 55° parallel, is the administrative capital of the Kativik Regional Government and the largest northern village in the Nunavik region of Québec, Canada (Figure 2a; [46]). The 2016 census indicated 2754 inhabitants [46]. Diorite and paragneiss are the main lithologies, but smaller occurrences of tonalite, gabbro and granite are also observed (Figure 2b). A general description of these units is given in SIGÉOM (Système d'information géominière du Québec) [47].

### 3.1. Kuujjuaq's Heating Demand

The community of Kuujjuaq experiences an annual average temperature of about 5.4 negative-degrees and an annual average of 8520 heating degree days below 18 °C (Figure 3; [15]). Although the residential dwellings are built to meet certain regulatory standards of insulation, the harsh climate results in high building heating requirements (Figure 3; [15]). The annual average fuel consumption of a typical residential dwelling in Kuujjuaq has been estimated as about 3100 to 8180 L [14,15]. This represents around 28 to 32 L m$^{-2}$, with respect to the floor area. There are currently about 973 occupied residential dwellings in Kuujjuaq [33], indicating a total yearly consumption of 3 to 8 million L of oil for space heating. The peak heating load for a residential dwelling is 7 kW [14] to 15 kW, depending on the building heating load and floor area, indicating a peak load for the community of approximately 7–15 MW. The annual heating energy demand has been estimated between 21.6 and 71.3 MWh per dwelling, depending on the floor area [14,15]. Thus, the community's heating demand is approximately 21 to 69 GWh per year.

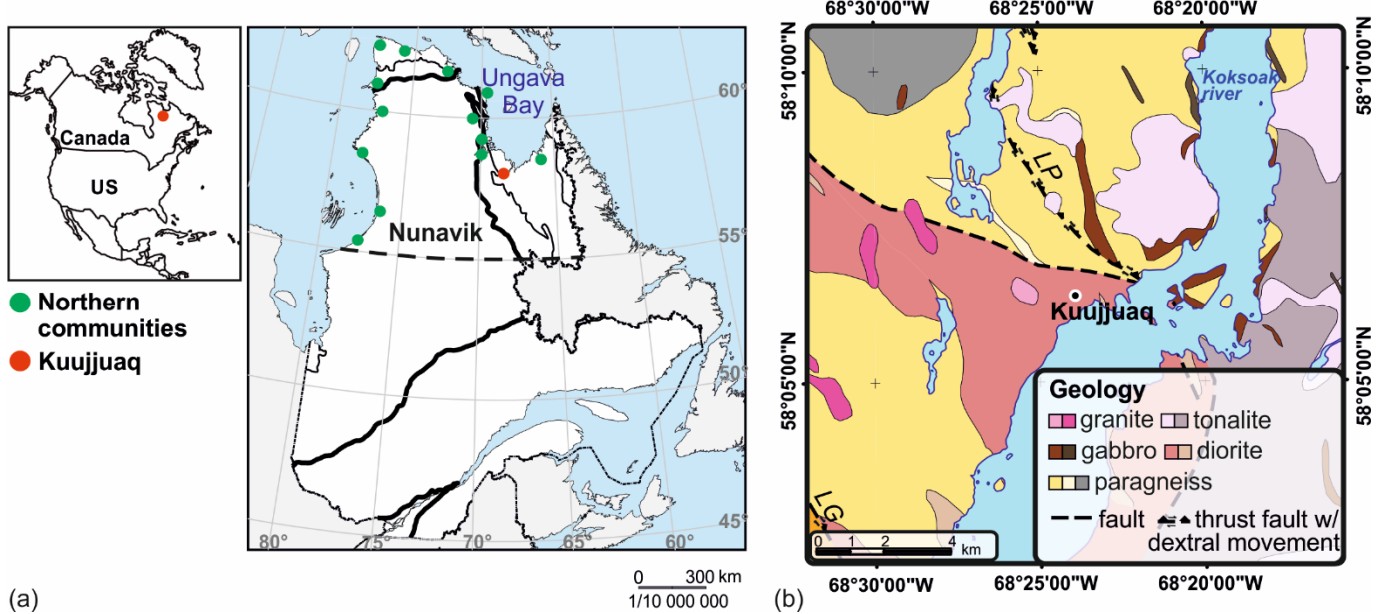

**Figure 2.** (**a**) Geographical location of Kuujjuaq and the remaining communities in Nunavik; (**b**) geological setting of the study area. *LP*—Lac Pingiajjulik fault, *LG*—Lac Gabriel fault. Adapted from SIGÉOM [47] and Miranda et al. [48].

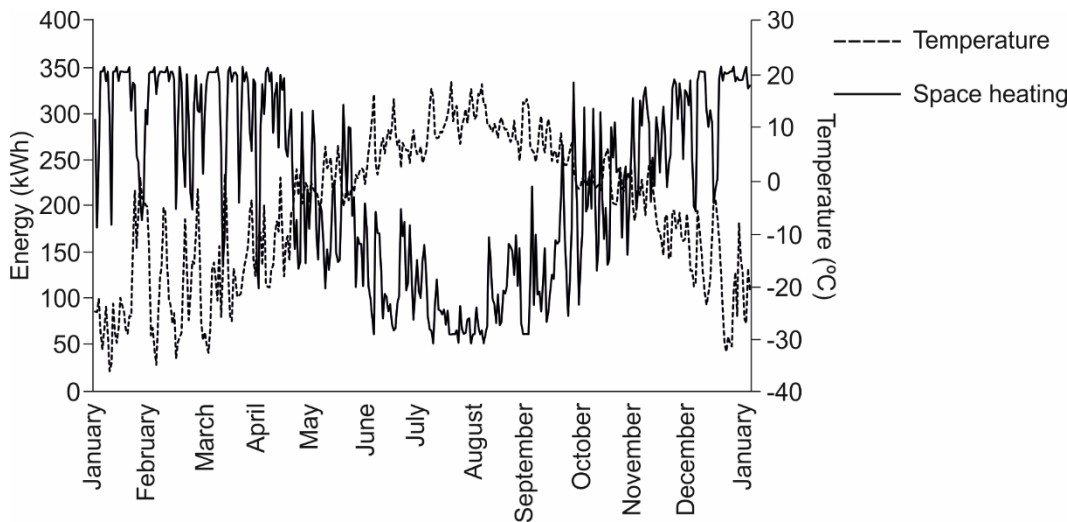

**Figure 3.** Average daily temperature and heating load profile of a typical residential dwelling in Kuujjuaq, redrawn from Gunawan et al. [15].

The population in Kuujjuaq grew by 16% between 2011 and 2016 [41], representing an annual growth of about 3%. Translating this population increase into heating energy needs, and carrying out projections, suggests that in 30 years, the community's annual heating demand may be 57 to 188 GWh. These values represent the threshold that the geothermal system designed in this study needs to meet during the 30 years of operation.

### 3.2. Previous Research Undertaken in Kuujjuaq

The lithologies outcropping in Kuujjuaq were sampled and a detailed description of texture, fabric, main mineral phases and major and trace geochemical elements is given in Miranda et al. [48]. Moreover, Miranda et al. [28,48] present the results of thermal conductivity, volumetric heat capacity, radiogenic heat production, density, porosity and primary rock permeability evaluated for the samples collected. The thermal conductivity and volumetric heat capacity were analyzed considering the samples at dry and water saturated states and for temperatures ranging between 20 and 160 °C [28].

Furthermore, a temperature profile was measured in a groundwater monitoring well located nearby the community and the terrestrial heat flux evaluated following a 1D inverse heat conduction approach, as explained in Miranda et al. [49] (Figure 4a). The heat flux assessment considered several hypotheses for the ground surface temperature history and variable conditions for the thermophysical properties. This approach suggested a heat flux at 10 km depth ranging between 32 and 69 mW m$^{-2}$. The evaluated subsurface temperature distribution for a depth up to 10 km considered these results and is presented in Miranda et al. [28]. At 4 km depth, the subsurface temperature was estimated to range between 28 °C and 167 °C, with a median of 88 °C, depending on the paleoclimate history and thermophysical properties conditions. The minimum and maximum temperature values were defined in a deterministic manner and may correspond to the least probable scenarios.

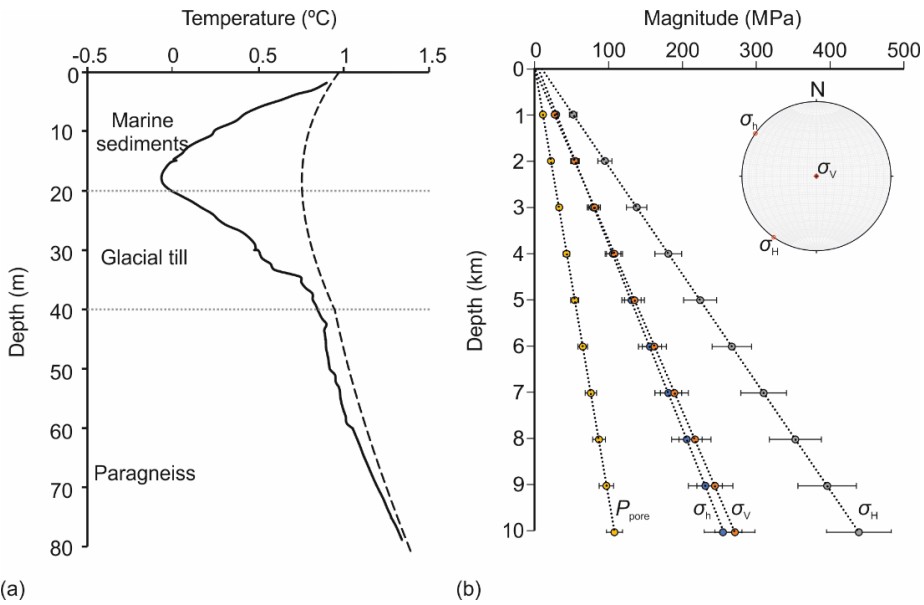

**Figure 4.** (**a**) Measured (solid line) and simulated (dashed line) temperature profile near the community of Kuujjuaq. The simulated profile corresponds to a heat flux of 41.6 mW m$^{-2}$ at 10 km depth (reprinted with permission from Miranda et al. [49], 2021 Elsevier Ltd.); (**b**) magnitude and orientation of the principal stresses. The error bar represents the range of values inferred. $\sigma_H$—maximum horizontal principal stress, $\sigma_h$—minimum horizontal principal stress, $\sigma_V$—vertical principal stress, $P_{pore}$—in situ fluid pressure.

Hypotheses for the specific discharge, Peclet number, hydraulic gradient and hydraulic conductivity are also discussed in Miranda et al. [49]. The specific discharge was calculated based on the hydraulic conductivity and hydraulic gradient. The permeability inferred for the paragneiss is $10^{-19}$ m$^2$, the hydraulic conductivity was evaluated as $10^{-13}$ m s$^{-1}$, the hydraulic gradient as $10^{-2}$, the specific discharge was inferred to be $10^{-14}$ m s$^{-1}$ and the Peclet number $10^{-6}$.

Miranda et al. [50] carried out a preliminary evaluation of the fracture network in terms of geometrical and topological properties that was further improved in Miranda et al. [51]. Four main fracture sets were identified but only one, the N-S set, is optimally oriented for slip. The sets E-W and NNW-SSE require higher additional fluid overpressure to be reactivated. Additionally, Miranda et al. [51] present an a priori stress estimation for Kuujjuaq calibrated with published stress data for the Canadian Shield and built upon geological indicators, empirical correlations, analytical models and Monte Carlo simulations to deal with the ensemble of uncertainties (Figure 4b). The orientation of the maximum horizontal stress was inferred N210-220°E. All of these previous results constitute the basis for the numerical simulations carried out in the present study.

## 4. Methods

### 4.1. Numerical Simulator

The numerical simulator used in this study is an updated version of Jing et al.'s [52] FRACSIM3D for new joint constitutive laws.

The model requires as input the stimulated rock volume and a given stimulation injection pressure to calculate the 3D shape of the stimulated volume, amounts of shear slip, fracture apertures and the amount of fluid required. This simple approach assumes that the stimulation proceeds as a "shock wave" with low fluid pressure gradients within the already stimulated volume and steep gradients at the outer margins [53]. Following stimulations(s), the steady state flow through the modified fracture system is calculated followed by tracer test stimulations and heat extraction. Multiple well segments, vertical or inclined, can be specified for stimulation and for injection and recovery.

A shear displacement–dilation relationship, in which the shear dilation angle is a reducing function of displacement rather than constant, was formulated and implemented. The rate of opening with displacement associated with this new sliding/opening law is in line with many experimental publications and geological observations, for example, the work of Lee and Cho [54].

Slip commences when the ratio of the shear stress to the normal stress exceeds a threshold. This ratio is expressed as the tangent of the total friction angle, which is a property of both rock material and geometry of the fracture surface, and it can be derived from tilted block experiments. The total friction angle is a function of the normal stress, the total displacement and the accumulated fracture surface damage from past slip movement. This angle is the sum of the basic friction angle of a smooth rock surface and the shear dilation angle, which is the arctangent of the ratio of the amount of fracture opening per small increment of displacement. The shear dilation angle can be derived from the roughness of the natural fracture surface, and the dilation with slip may be derived for various representations of surface roughness and correlation from one side of the fracture to the other. The fracture opening responds to increased effective normal stress by closure and the rate of closure is known as the fracture normal compliance. Theoretical and data driven forms for the closure curve can be derived for specific models of joint roughness and material properties (e.g., [54,55]); for our current purposes, much of the required observational data is unavailable so a simple form is desirable.

The Jing et al. [52] version of FRACSIM3D aims to capture shear dilation and normal compliance in a single equation as:

$$w = \frac{w_0 + U tan(\phi_{\text{dilation}})}{1 + 9\left(\frac{\sigma'_n}{\sigma_{n\_ref}}\right)} \tag{1}$$

where $w$ (m) is the aperture of the compliant fracture and $w_0$ (m) is the initial aperture before induced slip, $U$ (m) is the amount of slip (or shear displacement), $\sigma'_n$ (Pa) is the effective normal stress, $\sigma_{n\_ref}$ (Pa) is the reference stress for 90% closure (i.e., closure resistance of the fracture) and $\phi_{\text{dilation}}$ (°) is the shear dilation angle at low normal stress.

The shear dilation angle in the original Jing et al. [52] software version is assumed constant, meaning that a fracture tends to open at a constant rate as displacement increases at constant effective normal stress. Rock mechanics experiments (e.g., [54]) have suggested that the shear dilation angle increases over the first few millimeters of displacement as asperities of increasing wavelengths become out of phase, reaching a maximum and then decreasing to lower values after few millimeters of displacement since the longer wavelength asperities have less relative amplitude than those with the shorter wavelength. The assumption of a constant shear dilation angle was substituted by one that is a function of displacement to consider these observations. The relationship is as follows:

- A linear increase from a starting shear dilation angle value to a maximum value over a certain small shear displacement distance;

- An exponential decay with displacement thereafter to a low constant value at greater displacement at a user-specified rate.

Microseismic events recorded in the stimulation of crystalline rock may be caused by rapid changes in the shear dilation angle with displacement but rather caused by the sudden failure of macroscopic asperities or jogs. These asperities or jogs, not captured by the geometry of FRACSIM3D, might be related to fracture intersections, where upon the original propagation of one fracture is interrupted when it cuts another. The amount of shear stress needed to break these asperities is likely to be related to the asperity geometry, the number and extent of these asperities, the rock strength at the appropriate scale and the normal stress. To capture the possible effects of these macroasperities, an asperity strength factor was introduced into the code to give the fracture planes some extra resistance to slip. The asperity strength factor is converted to the rupturing shear stress required to overcome friction by:

$$\tau = F_{\text{asperity}} \sigma' n tan(\phi_{\text{basic}}) \tag{2}$$

where $\tau$ (Pa) is the shear stress, $F_{\text{asperity}}$ is the asperity strength factor and $\phi_{\text{basic}}$ (°) is the basic friction angle.

A brief iterative solution provides the fluid pressure in the fracture required to cause asperity rupture via a reduction in asperity strength through a reduction in effective normal stress. The value of the asperity strength factor is further varied about the mean to create a population of different fracture asperity strengths. The introduction of this factor allows the generation of microseismic events of an appropriate magnitude and numbers within the FRACSIM3D model. Given sufficient stimulation injection pressure, an asperity strength factor of 0.4 and above will generate large simulated microseismic events, while a value of zero will suppress the magnitude of microseismic events.

To calculate post-stimulation flow, the fractures generated are embedded within a regular cubic lattice 3D discretization grid, in which the quantity of fluid flow from block to block is controlled by Darcy's law with the transmissivity contribution from each fracture governed by the sum of products of the cubes of the fracture apertures and the fracture intersection length with the block face [52]:

$$Q_j = \sum_i \frac{w_{\text{h,i}}^3 l_i}{12\omega} \nabla P, \ j = x, y, z \tag{3}$$

where $Q_j$ (m$^3$ s$^{-1}$) is the flow rate, $w_\text{h}$ (m) is the fracture flow aperture, $l_i$ (m) is the fracture intersection length with the block face, $\omega$ (kg m$^{-1}$ s$^{-1}$) is the fluid dynamic viscosity and $\nabla P$ (Pa m$^{-1}$) is the pressure difference between blocks. Since the aperture of each fracture intersection with a lattice cube face depends on the local fluid pressure, the steady state flow solution is necessarily iterative. A simple over-relaxation scheme was chosen for this computationally intensive part of the simulation to be able to easily interrupt the convergence to update apertures as the solution progresses, to keep the code free of commercial library subroutines and to fit the solution within the confines of the computer memory available at the time the code was written.

In other words, the flow pattern for the discrete fracture network is solved by converting it to an equivalent continuum mesh (with a finer length scale than typical fractures that affect flow) where the steady state flow through fractures is expressed using the "cubic law" (i.e., Reynold's equation which is the solution of the Navier–Stokes equation for laminar flow between smooth parallel plates). Experiments (e.g., [54–59]) have suggested that the mechanical aperture of rough fractures overestimates the flow capacity due to the parallel plate assumption. Therefore, a flow aperture, somewhat less than the calculated mechanical aperture, is implemented. However, a user-definable input variable has been implemented to allow for manual input when desired.

Heat extraction is subsequently calculated under the approximate assumption that a thermal equilibrium is reached instantaneously between each solid element and the water passing through it. Moreover, heat transfer is constrained within the stimulated volume so

that no heat transfer occurs at the boundaries of the total model volume. Heat is transferred between elements by both conduction and advection [60]:

$$\rho c \frac{\partial T}{\partial t} = \nabla(\lambda \nabla T) - \rho c_{\text{fluid}} u \nabla T \tag{4}$$

where $\rho c$ (J m$^{-3}$ K$^{-1}$) is volumetric heat capacity, $T$ (°C) is temperature, $t$ (s) is time, $\lambda$ (W m$^{-1}$ K$^{-1}$) is thermal conductivity and $u$ (m s$^{-1}$) is Darcy velocity.

The thermal energy extracted by the circulating fluid is given by:

$$e_{\text{th}} = \rho c_{\text{fluid}} Q_{\text{recovered}} \left( T_{\text{recovered}} - T_{\text{injection}} \right) \tag{5}$$

where $e_{\text{th}}$ (W) is the thermal energy. The temperature in the reservoir varies over time as it cools. Further details on the joint constitutive laws, stimulation and steady state flow solutions, water loss approximation and heat extraction are given in Jing et al. [52].

*4.2. Model Geometry*

A cubic model volume of 4 km$^3$ (with 1.6 km of edge length) discretized into a grid of 200 per 200 per 200 cells is used to carry out the numerical simulations. This grid was selected after carrying out a grid dependency study and the results revealed an influence of, on average, 5%. Slippage takes place in the fractures whose centers fall within the current (gradually refined) estimate of the stimulated volume configuration. Although this volume is fixed and defined by the user, the shape of the reservoir is adjusted progressively as the estimate of the stimulated permeability tensor is refined. The major axis of the stimulated area generally becomes oriented according to the inferred direction of the maximum horizontal principal stress. The boundaries of the model are assumed to be connected to a constant pressure boundary a few kilometers beyond the model boundary and are able to leak off fluid or draw it in during the steady state flow calculation. For a more detailed explanation of the treatment of far field fluid losses/gains, see the discussion in Jing et al. [52]. The model boundaries were kept at a constant temperature since the heat extraction volume happens away from the boundaries. At time zero, the fractures within the model volume are assumed to be filled with the in situ fluid and the pressure distribution is assumed to be hydrostatic. The initial in situ rock mass permeability, from which the initial fracture apertures are calculated, and the state of stress are defined by the user. The far-field, well beyond the model boundaries, is assumed to be at hydrostatic pressure during the stimulation and circulation calculations.

The engineered geothermal energy system was designed as a doublet, with one injector and one producer, with vertical wells. Two configurations for the wells were studied to identify the best location relative to the Lac Pingiajjulik fault (Figure 5) and maximum horizontal principal stress assumed direction (NE-SW). Different possibilities for well spacing and open hole length were considered. The well bore radius was defined as 0.11 m following the example of Soultz's engineered geothermal energy system (e.g., [44]). The stimulation volume was considered variable. One stimulation was applied in each well and different stimulation and circulation pressures were studied. The operation time was defined for 30 years.

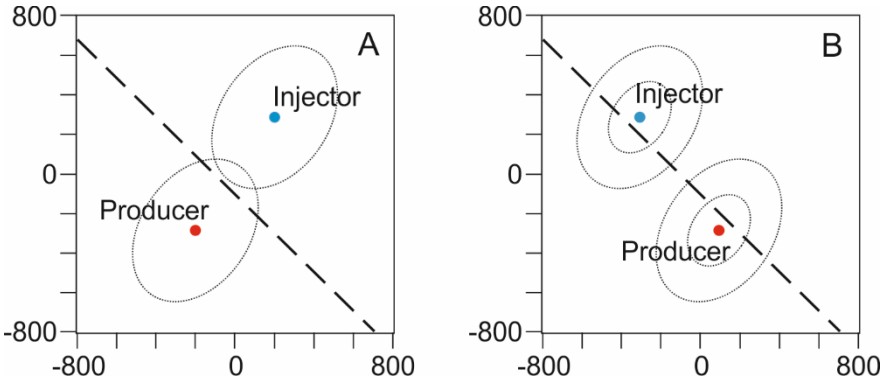

**Figure 5.** Top view slice cutting the center of the model and illustrating the different wells configuration studied. Dots—vertical wells, dashed line—Lac Pingiajjulik fault trace, ellipsoids—stimulation volume per well tried (larger ellipsoid, V = 0.4 km³; smaller ellipsoid, V = 0.2 km³; the sum of the stimulated volumes in each well were defined to correspond to 20% and 10%, respectively, of the total model volume).

Three working hypotheses for the fracture network were generated and studied due to the current existing uncertainties, as explained in Miranda et al. [51]. These are:

1. Fracture intensity of 0.8 fractures m$^{-1}$, fracture length and fractal dimension as sampled in the field (Figure 6a);
2. Fracture intensity of 0.8 fractures m$^{-1}$, fracture length increased by a factor of 10 and fractal dimension as sampled in the field (Figure 6b);
3. Fracture intensity of 0.8 fractures m$^{-1}$, fracture length as sampled in the field and fractal dimension decreased by a factor of 2 (Figure 6c).

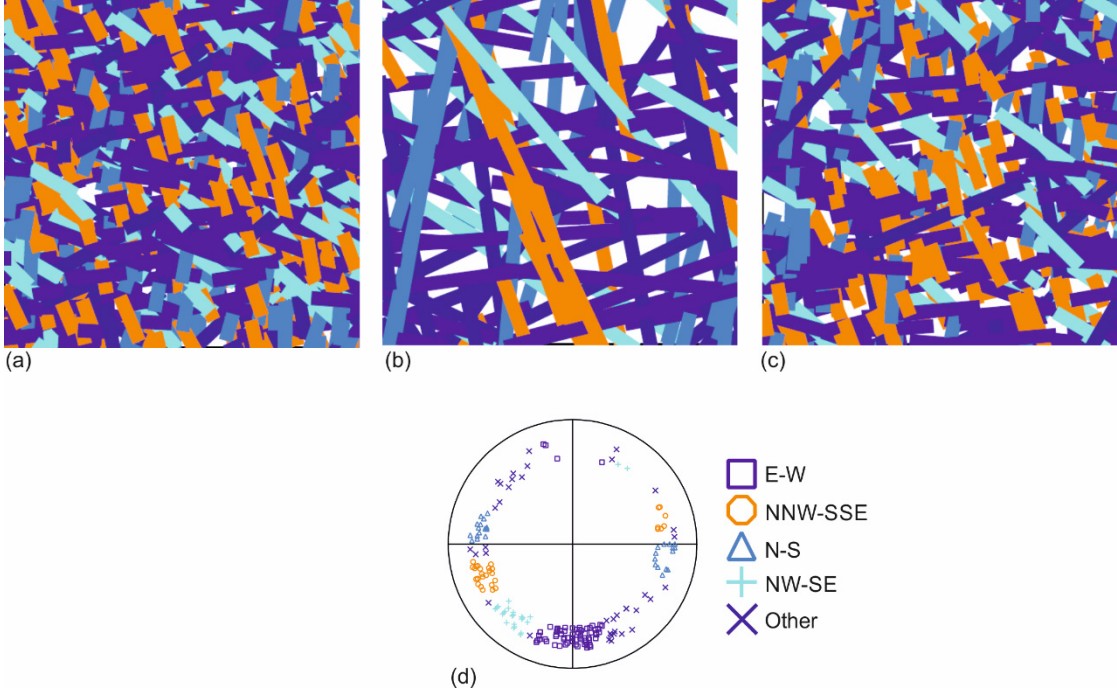

**Figure 6.** Top view of an 80 per 80 m section of the total generated fracture network model: (**a**) base-case fracture network; (**b**) fracture network with fracture lengths increased by a factor of 10; (**c**) fracture network with fractal dimension decreased by a factor of 2; (**d**) Wulff projection stereo plot of field fracture data. Adapted from Miranda et al. [51].

The influence of the Lac Pingiajjulik fault was assessed by adding the fault plane to the model in a deterministic manner. The fracture was assumed to cut approximately through the middle of the model (Figure 5). This structure is described as a thrust fault with late dextral movement (e.g., [61]). Thus, its dip was assumed 45°NE and its dip azimuth N45°E but its exact orientation at depth is unknown. The radius of this fault was assumed to be longer than 5 km and the initial fault aperture at in situ effective stress was considered variable between 0.001 and 0.10 m. These apertures in no way reflect the actual geological fault offset, but rather were simply chosen to generate different flow apertures for the fault structure (low and high). The Lac Pingiajjulik fault, considering the estimated stress field [51], has high effective stresses acting across it at the present day.

### 4.3. Properties of the Medium

The engineered geothermal energy system was designed for a depth of 4 km since previous analyses suggest a 98% probability that an adequate thermal energy source to meet the community's heating needs is available at this depth [28]. Preliminary simulations were carried out using a deterministic approach to define the worst-, base- and best-case scenarios in terms of properties of the medium (Table 1). It should be noted that the worst- and best-case scenarios may correspond to the least probable scenarios to occur since all of the best (and worst) values were grouped together.

**Table 1.** Sensitivity analysis for the properties of the medium.

| Parameter | Symbol | Unit | Worst-Case Scenario | Base-Case Scenario | Best-Case Scenario | References |
|---|---|---|---|---|---|---|
| Maximum horizontal principal stress | $\sigma_H$ | MPa | 213 | 181 | 152 | [51] |
| Direction maximum horizontal principal stress | $\sigma_H$ | ° | | N215°E | | [51] |
| Minimum horizontal principal stress | $\sigma_h$ | MPa | 138 | 106 | 84 | [51] |
| Direction minimum horizontal principal stress | $\sigma_h$ | ° | | N305°E | | [51] |
| Vertical principal stress | $\sigma_V$ | MPa | 121 | 108 | 97 | [51] |
| In situ fluid pressure | $P_{pore}$ | MPa | 49 | 43 | 39 | [51] |
| Reservoir temperature | $T_{reservoir}$ | °C | 33 | 88 | 167 | [28] |
| Permeability | $\kappa$ | m$^2$ | $10^{-18}$ | $10^{-17}$ | $10^{-16}$ | [62,63] |
| | | | Geological materials | | | |
| Thermal conductivity | $\lambda_{rock}$ | W m$^{-1}$ K$^{-1}$ | 2.4 | 2.0 | 1.5 | [28] |
| Volumetric heat capacity | $\rho c_{rock}$ | MJ m$^{-3}$ K$^{-1}$ | 2.1 | 2.4 | 2.7 | [28] |
| Young's modulus | $E$ | GPa | 100 | 71.5 | 43 | [64] |
| Poisson's ratio | $\nu$ | — | 0.16 | 0.23 | 0.30 | [64] |
| Asperity strength factor | $F_{asperity}$ | — | 0.6 | 0.5 | 0.4 | — |
| Basic friction angle | $\phi_{basic}$ | ° | 26 | 24.5 | 23 | [65] |
| Initial shear dilation angle | $\phi_{dilation, 0}$ | ° | 0 | 2.5 | 5 | — |
| Peak shear dilation angle | $\phi_{dilation, peak}$ | ° | 5.0 | 10 | 20 | — |
| Ultimate shear dilation angle | $\phi_{dilation, ultimate}$ | ° | 2.5 | 5.0 | 10 | — |
| Peak shear displacement | $U_{peak}$ | mm | 2.5 | 5.0 | 10 | — |
| Residual shear displacement | $U_{residual}$ | mm | 1.25 | 2.5 | 5 | — |
| Reference stress for 90% closure | $\sigma_{n, ref}$ | MPa | 40 | 50 | 60 | — |

**Table 1.** *Cont.*

| Parameter | Symbol | Unit | Worst-Case Scenario | Base-Case Scenario | Best-Case Scenario | References |
|---|---|---|---|---|---|---|
| | | In situ fluid | | | | |
| Density | $\rho_{\text{fluid}}$ | kg m$^{-3}$ | 1012 | 1080 | 1112 | — |
| Dynamic viscosity | $\omega$ | kg m$^{-1}$ s$^{-1}$ | $7.48 \times 10^{-4}$ | $3.19 \times 10^{-4}$ | $1.62 \times 10^{-4}$ | — |
| | | Circulation fluid | | | | |
| Re-injection temperature | $T_{\text{injection}}$ | °C | 30 | 30 | 50 | — |
| Density | $\rho_{\text{fluid}}$ | kg m$^{-3}$ | 993 | 993 | 985 | — |
| Specific heat | $c_{\text{fluid}}$ | J kg$^{-1}$ K$^{-1}$ | 4180 | 4180 | 4180 | — |

The magnitude of the principal stresses and in situ fluid pressure were inferred by Miranda et al. [51] (Figure 4b) using empirical correlations, analytical modeling and Monte Carlo simulations calibrated with published literature data for the Canadian Shield. The subsurface temperature distribution was inferred in Miranda et al. [28]. Hypotheses for the in situ permeability are based on the results obtained at the Rosemanowes and Soultz stimulated geothermal projects [62,63]. The thermal conductivity is a function of temperature and pressure and was evaluated by Miranda et al. [28]. The volumetric heat capacity for water-saturated samples has been evaluated by Miranda et al. [28]. The Young's modulus and Poisson's ratio were defined based on the findings of Arjang and Herget [64] for the Canadian Shield. Asperity strength factor, initial, peak and ultimate shear dilation angle, peak and residual shear displacement and reference stress for 90% closure are working hypotheses to evaluate their influence on the performance of the hydraulically stimulated geothermal reservoir. It is important to highlight that these values are working hypotheses only and may be subject to wide variation. The basic friction angle was defined according to literature tilt experiments carried out in wet crystalline rock masses (e.g., [65]). The density of both in situ and circulation fluids was calculated as ([66] and references therein):

$$\rho_{\text{fluid}}(P, T) = \rho_{\text{fluid}}(T) + \Delta\rho_{\text{fluid}}(P) + 1000 \times TDS \tag{6}$$

where *TDS* (kg L$^{-1}$) is the total dissolved solids and *P* (Pa) stands for pressure.

The effect of temperature and pressure are given by ([66] and references therein):

$$\rho_{\text{fluid}}(T) = \frac{999.84 + 16.95T + \left(-7.99 \times 10^{-3}\right)T^2 + \left(-4.62 \times 10^{-5}\right)T^3 + \left(1.06 \times 10^{-7}\right)T^4 + \left(-2.81 \times 10^{-10}\right)T^5}{1 + 0.017T} \tag{7a}$$

$$\Delta\rho_{\text{fluid}}(P) = \left(4.0625 \times 10^{-7}\right)P \tag{7b}$$

The in situ fluid was assumed brine with 100 g L$^{-1}$ of total dissolved solids [67] and in thermal equilibrium with the geological medium. The dynamic viscosity of the in situ fluid was approximated as [66]:

$$\omega(T) = \left(2.414 \times 10^{-5}\right) \times 10^{\frac{247.8}{T-140}} \tag{8}$$

where *T* (K) is the absolute temperature.

Preliminary calculations were carried out to assess a baseline flow rate that the simulations need to meet. This was done as:

$$Q = \frac{e}{\rho c_{\text{fluid}}\left(T_{\text{reservoir}} - T_{\text{injection}}\right)} \tag{9}$$

where *e* (W) is the maximum heating energy demand.

The results suggest that flow rates in the range 14 to 45 L s$^{-1}$ are required for the best-case scenario while for the base-case scenario, the flow rate range necessary is 27 to

89 L s$^{-1}$. For the worst-case scenario, in turn, flow rates above 200 L s$^{-1}$ are needed. The flow rates estimated for the best- and base-case scenarios have already been achieved in engineered geothermal energy systems (e.g., Soultz and Rittershoffen; [68]).

### 4.4. Levelized Cost of Energy

A first-order evaluation of the levelized cost of energy to build and operate an engineered geothermal energy system in Kuujjuaq at a depth of 4 km was carried out as (e.g., [69]):

$$\text{LCOE} = \frac{A_{\text{total}}}{e_{\text{annual}}} = \frac{O_{\text{annual}} + J_{\text{annual}}}{e_{\text{annual}}} = \frac{O_{\text{annual}} + \frac{i(1+i)^t}{(1+i)^t-1}I_{\text{total}}}{e_{\text{annual}}} \tag{10}$$

where LCOE ($ MWh$^{-1}$) is the levelized cost of energy, $O_{\text{annual}}$ (¢ kWh$^{-1}$) is the annual operation and maintenance cost, $e_{\text{annual}}$ (MWh) is the energy provided annually, $i$ (%) is the imputed interest rate, $t$ (year) is the project lifetime and $I_{\text{total}}$ ($) is the total capital investment. The latter is estimated as (e.g., [70]):

$$I_{\text{total}} = C_{\text{wells}} + C_{\text{stimulation}} + C_{\text{plant}} \tag{11}$$

where $C$ ($) stands for cost.

Three different well cost models were used. These are (e.g., [71] and references therein):

$$\text{WellCost Lite} = F_s \times 10^{-0.67+0.000334(z+h)} \times N \tag{12a}$$

$$\text{ThermoGIS} = F_s \times \left[0.2(z+l)^2 + 700(z+l) + 25000\right] \times N \times 10^{-6} \tag{12b}$$

$$\text{HSD} = 1500 \times z \times N \times 10^{-6} \tag{12c}$$

where $F_s$ is the well cost scaling factor, $z$ (m) is depth, $l$ (m) is the possible extra horizontal length of the well and $N$ is the number of wells. HSD stands for hydrothermal spallation drilling. The parameter $F_s$ was assumed 1 and $l$ was assumed 0 in this work.

The imputed interest rate, stimulation, power plant and other surface facilities and operation and maintenance costs were assumed the same as the values proposed in Sanyal et al. [72]. However, these costs may be underestimated for Kuujjuaq. In fact, due to the remoteness, the infrastructure cost in Nunavik can be 2 to 5 times more expensive than the regular construction/infrastructure costs in southern areas [73]. Therefore, a first analysis was carried out using Sanyal et al. [72] costs and these were posteriorly increased by a factor ranging between 2 and 5 to assess more realistic energy costs for Kuujjuaq (Table 2).

**Table 2.** Sensitivity analysis for the costs.

| Cost | | Sanyal et al. [72] (Optimistic Scenario) | Factor of 2 (Likely Scenario) | Factor of 5 (Pessimistic Scenario) |
|---|---|---|---|---|
| Stimulation per well (M$) | Minimum | 0.5 | 1.0 | 2.5 |
| | Mean | 0.75 | 1.5 | 3.75 |
| | Maximum | 1.0 | 2.0 | 5.0 |
| Power plant and other surface facilities ($ kWh$^{-1}$) | Minimum | 1800 | 3600 | 9000 |
| | Mean | 2000 | 4000 | 10,000 |
| | Maximum | 2200 | 4400 | 11,000 |
| Annual operation and maintenance (¢ kWh$^{-1}$) | Minimum | 2.0 | 4.0 | 10.0 |
| | Maximum | 3.5 | 7.0 | 17.5 |

The costs are in USD.

The imputed interest rate was assumed 9% [72]. The annually provided energy was estimated based on the annual average heating (and electricity) consumption inferred for the community. The ensemble of uncertainties was simulated with the Monte Carlo

method (Table 3) using the software @RISK [74] and the simulations were carried out for each possible engineered geothermal energy system design computed with FRACSIM3D.

**Table 3.** Levelized cost of energy and uncertainty.

| Parameter Code | Parameter Description | Unit | Variable Type | Distribution |
|:---:|:---:|:---:|:---:|:---:|
| $C_{wells}$ | Wells cost | \$ | Discrete | Uniform |
| $C_{stimulation}$ | Stimulation cost | \$ | Continuous | Triang (min, mean, max) |
| $C_{plant}$ | Power plant and other surface facilities cost | $\$ \, kW^{-1}$ | Continuous | Triang (min, mean, max) |
| $O_{annual}$ | Annual operations and maintenance cost | $¢ \, kWh^{-1}$ | Continuous | Uniform (min, max) |
| $e_{annual}$ | Annually consumed energy | MWh | Continuous | Uniform (min, max) |
| $i$ | Imputed interest rate | % | Continuous | Single value |
| $t$ | Project lifetime | year | Continuous | Single value |

## 5. Results

The numerical simulations carried out aimed at: (1) studying how the fracture network and properties of the medium influence the initial fracture aperture, fracture shear stiffness and fracture shear displacement, (2) analyze how the stimulation fluid pressure influences the number of sheared fractures and (3) designing an engineered geothermal energy system capable to meet the following targets:

- Flow rates able to harvest enough thermal energy during all system operation time to meet the community's heating demand;
- Water loss maximum 20%;
- Reservoir flow impedance lower than $1 \, MPa \, L^{-1} \, s^{-1}$;
- Thermal drawdown lower than $1 \, °C/year$.

### 5.1. Initial Fracture Aperture

The initial fracture aperture is estimated in FRACSIM3D through a predictive model that considers the measured or estimated average permeability of the undisturbed rock mass, the fracture radius, density, orientation and in situ stresses [52,75,76]. The results reveal that the hydromechanical properties significantly influence the estimated initial aperture of the fractures (Figure 7). The size of the fractures and fractal dimension also play a minor role (Figure 7). The observed variability consequently influences the hydraulic conductivity of the medium and the performance of the hydraulic stimulation procedure. These results were computed assuming the wells in configuration A (Figure 5).

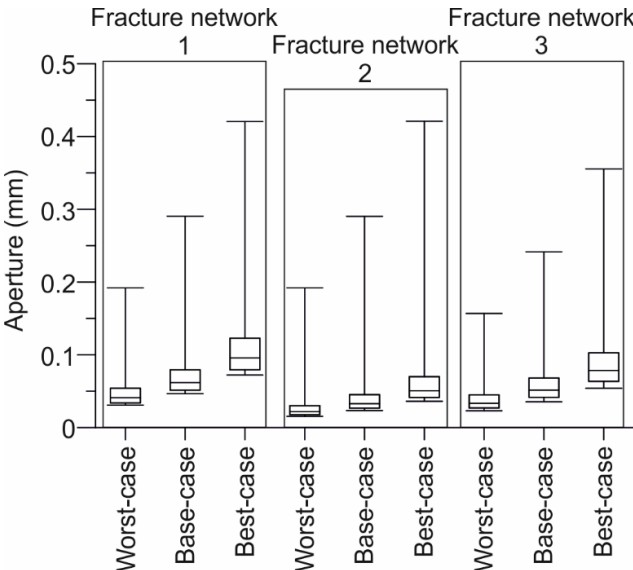

**Figure 7.** Boxplot of initial fracture aperture with whiskers from minimum to maximum. The reader is referred to Section 4.2 for further details on the fracture network and to Table 1 for further information on the properties of the medium.

### 5.2. Fracture Shear Stiffness

The fracture shear stiffness is approximated in FRACSIM3D as the ratio of the rock shear modulus divided by the fracture radius and multiplied by a geometric parameter [52,77]. Longer fractures, small Young's modulus and high Poisson's ratio are shown to considerably reduce the fracture shear stiffness of the medium (Figure 8), thus improving the performance of the hydraulic stimulation procedure. These results were computed assuming the wells in configuration A (Figure 5).

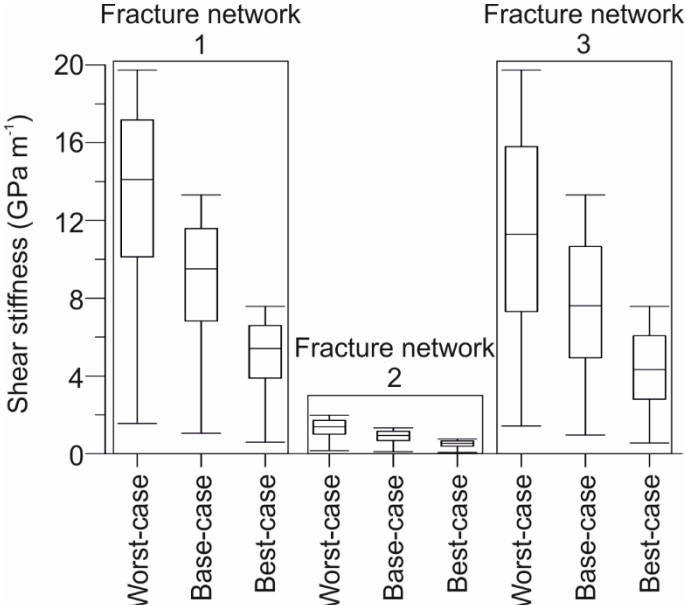

**Figure 8.** Boxplot of fracture shear stiffness with whiskers from minimum to maximum. The reader is referred to Section 4.2 for further details on the fracture network and to Table 1 for further information on the properties of the medium.

### 5.3. Fracture Shear Displacement

The shear displacement is expressed in FRACSIM3D as the excess shear stress over the required for slip divided by the fracture shear stiffness [52]. Thus, fractures with high shear stiffness tend to experience lower shear displacements and the latter tends to increase as a function of the applied fluid pressure (Figure 9). For the worst-case scenario, slippage occurs at stimulation, or fluid, pressures between 40 MPa and 70 MPa, regardless of the fracture network. For the base-case scenario, slip takes place at stimulation pressures of 20 to 60 MPa while, for the best-case scenario, between the pressures of 5 and 40 MPa. It is convenient to indicate that these results were computed assuming the wells in configuration A (Figure 5).

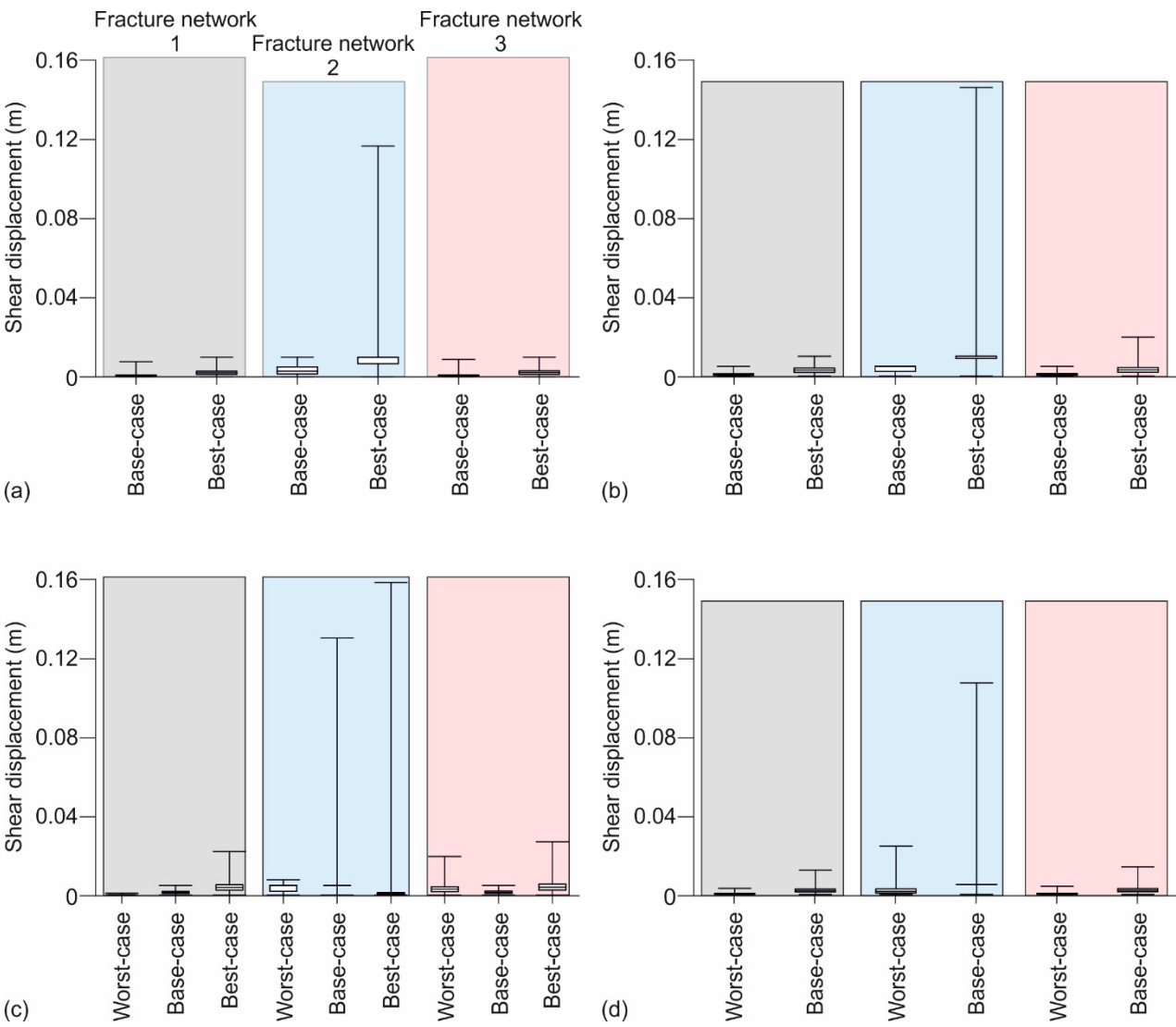

**Figure 9.** Boxplot of fracture shear displacement with whiskers from minimum to maximum: (**a**) stimulation pressure of 20 MPa; (**b**) stimulation pressure of 30 MPa; (**c**) stimulation pressure of 40 MPa; (**d**) stimulation pressure of 50 MPa. The reader is referred to Section 4.2 for further details on the fracture network and to Table 1 for further information on the properties of the medium.

### 5.4. Sheared Fractures

The optimally oriented fractures to slip belong to the sets E-W and N-S, followed by the set NNW-SSE and the fractures within the other group (Figure 10). The set NW-SE is not optimally oriented to slip (Figure 10). These observations confirm the Mohr–Coulomb

friction and slip tendency analyses carried out by Miranda et al. [51]. Moreover, increasing the stimulation fluid pressure led to an increase on the number of fractures that can be reactivated. However, it is important to highlight that the stress regime plays a major role and large uncertainty still exists as discussed by Miranda et al. [51]. As a note, these results were computed assuming the wells in configuration A (Figure 5) and a friction coefficient of 0.6.

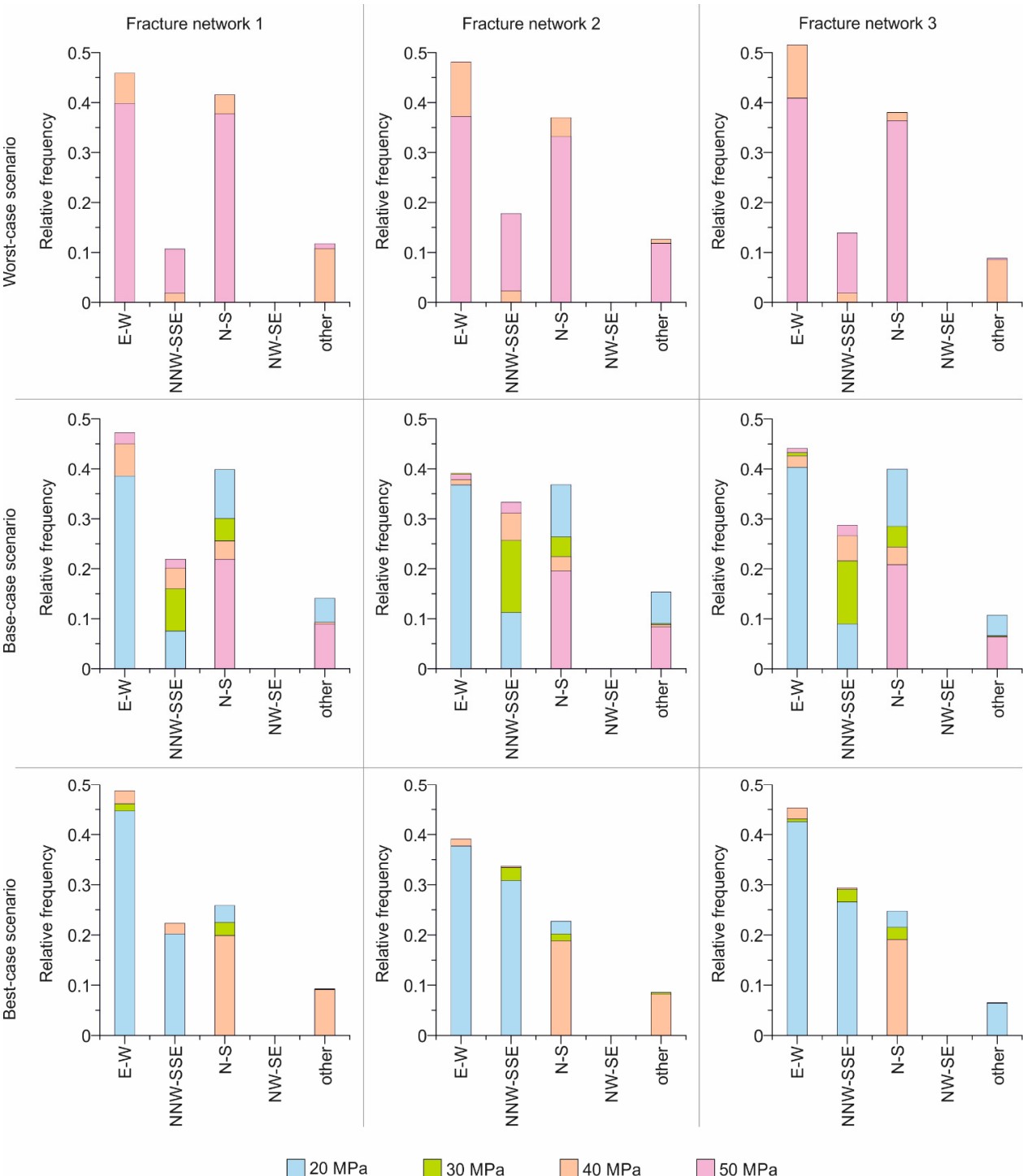

**Figure 10.** Sheared fractures grouped by sets for different stimulation fluid pressure. The reader is referred to Section 4.2 for further details on the fracture network and to Table 1 for further information on the properties of the medium. Relative frequency, in this context, corresponds to the percentage of fractures that slipped in each set from the total of fractures that slip.

### 5.5. Engineered Geothermal Energy System Design

The previous results suggest that developing a hydraulically stimulated geothermal reservoir is favored in a mechanically weak (i.e., low magnitude for the principal stresses and fractures with low resistance to deformation and opening) and hydraulically conductive medium. Moreover, they also indicate that smaller fractures tend to slip less, thus, making fluid circulation more difficult through these fractures. Reservoirs of potential interest were not simulated for the worst-case scenario, primarily due to the low reservoir temperature and high flow rates required to meet the demand, and secondly due to the high magnitude inferred for the principal stresses and high resistance of the fractures to slip and opening. Thus, the following sections will be focused on the best- and base-case scenarios, discussing the best simulated design to meet the water loss, reservoir impedance and thermal drawdown targets. The influence of the Lac Pingiajjulik fault and its initial aperture and, thus, hydraulic conductivity, is also considered.

#### 5.5.1. Best-Case Scenario

Reservoirs capable of meeting the aforementioned targets were simulated in this scenario for the three fracture network hypotheses regardless of Lac Pingiajjulik initial fault aperture (Table 4). As previously outlined, these simulations were carried out bearing in mind the estimated required flow rate of 14 to 45 L s$^{-1}$. The simulations reveal that, in this case study, positioning the wells perpendicular to the fault trace (configuration A; Figure 5) leads to a better performance and sustainability of the system than placing the wells parallel to the fault trace (configuration B; Figure 5). The results also suggest that longer fractures (fracture network 2) tend to improve the performance of the engineered geothermal energy system. Lower stimulation and circulation pressures are required to enhance the transmissivity and circulate the fluid.

#### 5.5.2. Base-Case Scenario

As previously outlined, these simulations were carried out bearing in mind the estimated required flow rate of 27 to 89 L s$^{-1}$. Reservoirs of potential interest were only generated for fractures longer than sampled in the field (fracture network 2). However, to meet the defined targets, higher stimulation volume and stimulation and circulation pressures were required than for the best scenario previously studied (Table 5).

**Table 4.** Design parameters and results.

| Configuration [1] | Fault Offset (m) | Spacing between Wells (m) | Open Hole Length (m) | $V_{stim}$ (km³) | $P_{stim}$ (MPa) $I^*$ | $R^*$ | $P_{circ}$ (MPa) $I^*$ | $R^*$ | $Q_{rec}$ (L s⁻¹) | $W_{loss}$ (%) | $H$ (MPa L⁻¹ s⁻¹) | $T_{drawdown}$ (°C/year) |
|---|---|---|---|---|---|---|---|---|---|---|---|---|
| | | | | | Fracture network 1 | | | | | | | |
| A | 0.10 | 700 | 600 | 0.8 | 25 | 28 | 10 | −5 | 52.9 | 18.1 | 0.28 | 0.16 |
| A | 0.01 | 700 | 600 | 0.8 | 25 | 28 | 9 | −5 | 47.4 | 17.6 | 0.30 | 0.12 |
| A | 0.001 | 500 | 600 | 0.8 | 29 | 35 | 6 | −4 | 49.0 | 15.5 | 0.20 | 0.10 |
| B | 0.10 | 700 | 500 | 0.4 | 23 | 28 | 7 | −5 | 22.4 | 20.4 | 0.54 | 0.60 |
| B | 0.01 | 700 | 500 | 0.4 | 25 | 30 | 7 | −5 | 28.8 | 18.2 | 0.42 | 0.73 |
| | | | | | Fracture network 2 | | | | | | | |
| A | 0.10 | 700 | 600 | 0.8 | 7 | 14 | 2 | 0 | 55.6 | 13.7 | 0.04 | 0.03 |
| A | 0.01 | 700 | 600 | 0.8 | 4 | 9 | 2 | −1 | 56.4 | 2.1 | 0.05 | 0.03 |
| A | 0.001 | 600 | 600 | 0.8 | 5 | 10 | 2 | −1 | 50.1 | 5.3 | 0.06 | 0.02 |
| B | 0.10 | 700 | 600 | 0.4 | 2 | 4 | 1 | −1 | 22.8 | 11.8 | 0.09 | 0.50 |
| B | 0.01 | 700 | 600 | 0.4 | 3 | 5 | 1 | −1 | 26.1 | 3.4 | 0.08 | 0.35 |
| | | | | | Fracture network 3 | | | | | | | |
| A | 0.10 | 700 | 600 | 0.8 | 23 | 24 | 8 | −4 | 50.7 | 8.0 | 0.24 | 0.07 |
| A | 0.01 | 600 | 600 | 0.8 | 23 | 24 | 6 | −4 | 46.8 | 9.9 | 0.21 | 0.05 |
| A | 0.001 | 500 | 600 | 0.8 | 26 | 32 | 6 | −4 | 51.8 | 5.0 | 0.19 | 0.04 |
| B | 0.10 | 700 | 600 | 0.4 | 22 | 24 | 3 | −3 | 22.3 | 2.9 | 0.27 | 0.45 |
| B | 0.01 | 700 | 600 | 0.4 | 22 | 24 | 4 | −3 | 22.9 | 20.5 | 0.31 | 0.56 |

[1.] The reader is referred to Figure 4 for further details on the configuration. $V_{stim}$—stimulation volume, $P_{stim}$—stimulation pressure, $P_{circ}$—circulation pressure, $I^*$—injection well, $R^*$—recovery well, $Q_{rec}$—recovered flow rate, $W_{loss}$—water loss, $H$—hydraulic impedance, $T_{drawdown}$—temperature drawdown. The reader is referred to Section 4.2 for further details on the fracture network.

**Table 5.** Design parameters and results.

| Configuration [1] | Fault Offset (m) | Spacing between Wells (m) | Open Hole Length (m) | $V_{stim}$ (km³) | $P_{stim}$ (MPa) $I^*$ | $R^*$ | $P_{circ}$ (MPa) $I^*$ | $R^*$ | $Q_{rec}$ (L s⁻¹) | $W_{loss}$ (%) | $H$ (MPa L⁻¹ s⁻¹) | $T_{drawdown}$ (°C/year) |
|---|---|---|---|---|---|---|---|---|---|---|---|---|
| | | | | | Fracture network 2 | | | | | | | |
| A | 0.10 | 700 | 600 | 0.8 | 47 | 44 | 14 | −2 | 93.3 | 0.4 | 0.17 | 0.47 |
| A | 0.01 | 700 | 600 | 0.8 | 47 | 44 | 16 | −4 | 96.6 | 10.6 | 0.21 | 0.58 |
| A | 0.001 | 700 | 600 | 0.8 | 48 | 45 | 16 | −4 | 88.2 | 18.2 | 0.23 | 0.90 |
| B | 0.10 | 700 | 600 | 0.8 | 43 | 45 | 3 | −1 | 29.8 | 9.7 | 0.13 | 0.87 |
| B | 0.01 | 700 | 600 | 0.8 | 43 | 45 | 4 | −2 | 32.2 | 2.7 | 0.19 | 0.73 |

[1.] The reader is referred to Figure 4 for further details on the configuration. $V_{stim}$—stimulation volume, $P_{stim}$—stimulation pressure, $P_{circ}$—circulation pressure, $I^*$—injection well, $R^*$—recovery well, $Q_{rec}$—recovered flow rate, $W_{loss}$—water loss, $H$—hydraulic impedance, $T_{drawdown}$—temperature drawdown. The reader is referred to Section 4.2 for further details on the fracture network.

### 5.6. Thermal Energy and Potential Heat Output

The thermal energy extracted by each design previously studied is capable of fulfilling the forecast heating energy demand over 30 years of operation (Figure 11). Moreover, the results indicate that configuration A (Figure 5) is more appropriate to meet the upper bound of the projected demand while configuration B (Figure 5) would more likely meet the lower bound of the demand.

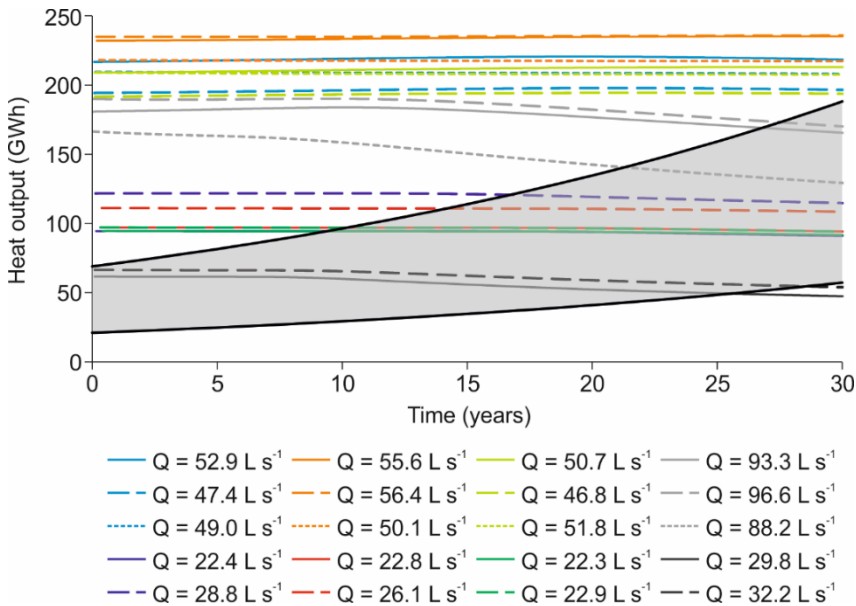

**Figure 11.** Heating energy produced (color lines) and projected demand (grey polygon). Lower bound of grey polygon—annual heating energy demand estimated based on Yan et al. [14], upper bound of grey polygon—annual heating energy demand estimated based on Gunawan et al. [15]. The reader is referred to Tables 4 and 5 for further details on the flow rates.

Some of the designs largely exceed the heating needs within the first years of the geothermal system operation. Therefore, an additional study was undertaken to assess if the geothermal system can be used for combined heat and power to improve the sustainability of the system. The electricity consumption in Kuujjuaq was evaluated as 18.9 GWh (see [28] for further details) and the yearly growth in energy demand is assumed to be 3% [46]. The results suggest that about 6 to 8 $MW_{th}$ are rejected per each $MW_e$ produced when considering the deep geothermal energy sources in Kuujjuaq. If only 50% of the waste heat from power production can be used for space heating, since the remaining is lost from parasitic equipment, an engineered geothermal energy system can be designed to operate in combined heat and power mode during approximately 8–15 years of the project lifetime, meeting both power and heating demand (Figure 12).

### 5.7. Recovery Factor

The thermally active reservoir volume changes over time as heat is being extracted. The recovery factor was calculated as the ratio between the thermally active volume of rock during the 30 years of operation and the defined stimulated volume (Figure 13). The recovery factor is commonly assumed to be within the theoretical range of 2–20% (e.g., [35,78]). The results obtained agree with these values.

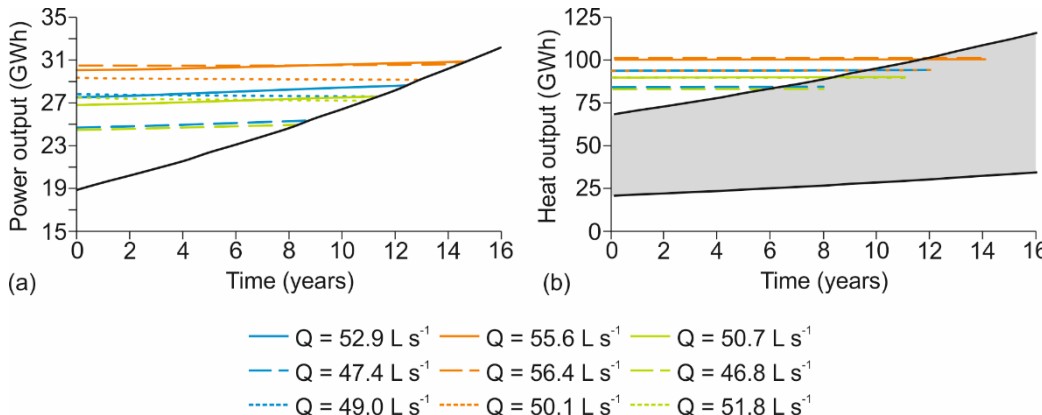

**Figure 12.** (**a**) Power and (**b**) heating energy produced from the waste heat (color lines) and projected demand (black line and grey polygon). Lower bound of grey polygon—annual heating energy demand estimated based on Yan et al. [14], upper bound of grey polygon—annual heating energy demand estimated based on Gunawan et al. [15]. The reader is referred to Tables 4 and 5 for further details on the flow rates.

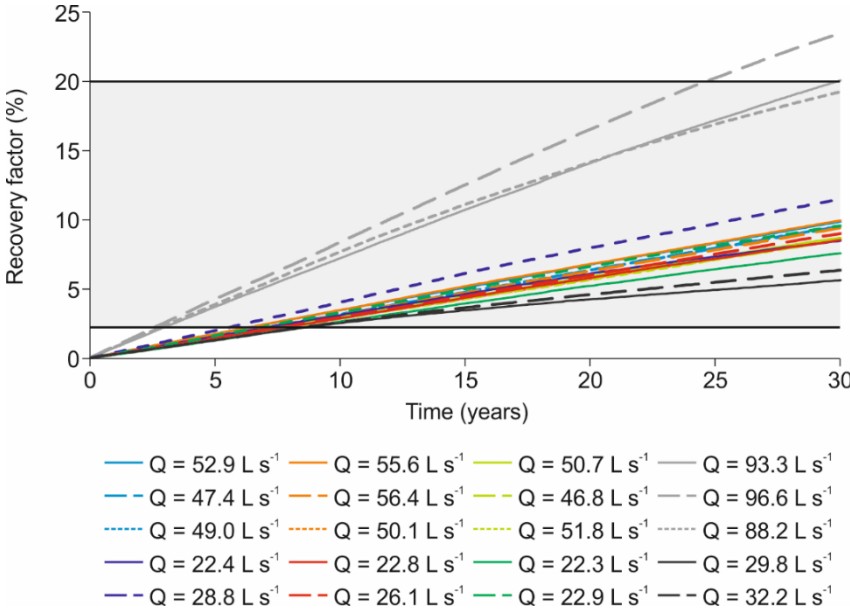

**Figure 13.** Recovery factor over time. Grey polygon—theoretical values for the recovery factor, lower bound = 2%, upper bound = 20% (e.g., [35,78]). The reader is referred to Tables 4 and 5 for further details on the flow rates.

### 5.8. Levelized Cost of Energy

The three well cost models used in this work suggest that the price of two 4-km wells can be USD 9.8 M (WellCost Lite model; Equation (12a)) and USD 12 M (Thermo-GIS and HSD models; Equation (12b) and Equation (12c), respectively). Considering the Sanyal et al. [72] proposed costs, the stimulation of both of the wells ranges between a minimum of USD 1 M and a maximum of USD 2 M, with a mean value of USD 1.5 M. The power plant and other surface facilities estimated for each possible design previously studied (Tables 4 and 5) is found to vary between an average maximum cost, for configuration A (Figure 5), of USD 54 M and an average minimum of USD 34 M (Figure 14a). For configuration B (Figure 5), the cost varies between an average minimum of USD 12 M and an average maximum of USD 28 M (Figure 14b). Thus, the average minimum capital cost estimated for configuration A (Figure 5) is USD 46 M while the average maximum is USD 66.8 M

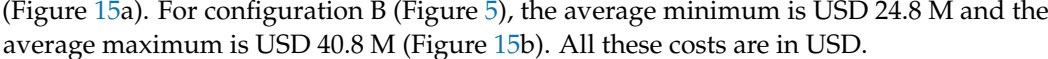

(Figure 15a). For configuration B (Figure 5), the average minimum is USD 24.8 M and the average maximum is USD 40.8 M (Figure 15b). All these costs are in USD.

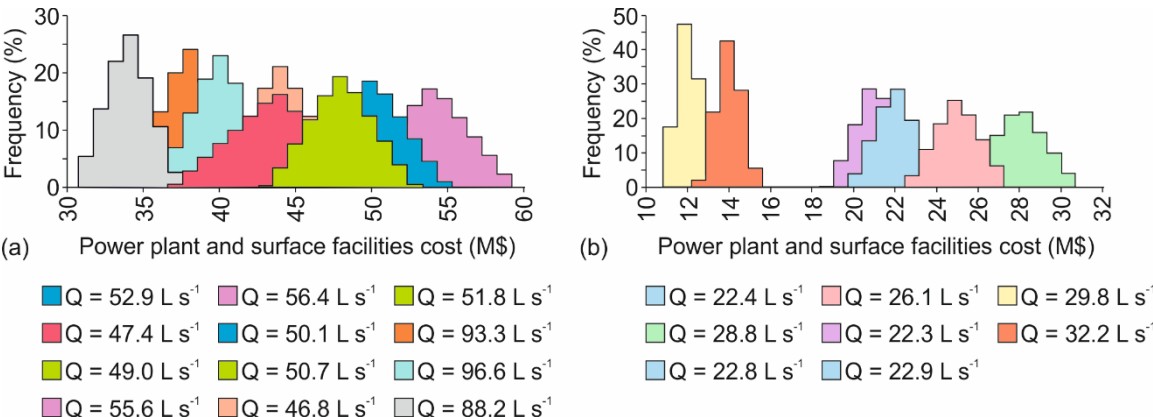

**Figure 14.** Power plant and other surface facilities cost for each simulated design considering Sanyal et al. [72] costs: (**a**) configuration A (Figure 5); (**b**) configuration B (Figure 5). The cost is in USD. The reader is referred to Tables 4 and 5 for further details on the flow rates.

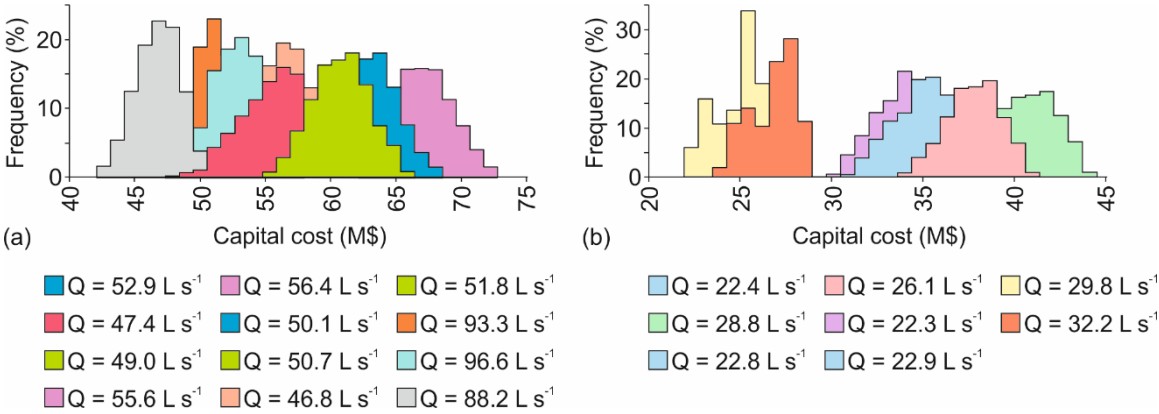

**Figure 15.** Capital cost for each simulated design considering Sanyal et al. [72] costs: (**a**) configuration A (Figure 5); (**b**) configuration B (Figure 5). The cost is in USD. The reader is referred to Tables 4 and 5 for further details on the flow rates.

However, these costs may be too optimistic since, due to the remoteness, the infrastructure cost in Nunavik can be 2 to 5 times more expensive than the regular construction/infrastructure costs in southern areas. Therefore, a second analysis was carried out increasing the costs by a factor of 2 and 5 (Table 2). The stimulation of both wells is found to vary between USD 2 M and USD 4 M, for a factor of 2, and between USD 5 M and USD 10 M if the costs are increased by 5 times. The power plant and other surface facilities, for a factor of 2, can have a cost varying within the average minimum of USD 24 M to the average maximum of USD 108 M (Table 6) or between USD 60 M and USD 270 M (Table 6) if the cost is 5 times higher than the one proposed by Sanyal et al. [72].

Consequently, the capital cost for each simulated design increases to an average minimum of USD 38 M and an average maximum of USD 122 M if the costs are 2 times higher and to USD 79 M and USD 289 M if the costs are 5 times higher than in Sanyal et al. [72] (Table 7).

**Table 6.** Power plant and other surface facilities cost for each simulated design.

| Design | | Power Plant and Other Surface Facilities Cost (M$) | | | | | |
|---|---|---|---|---|---|---|---|
| | | Factor of 2 (Likely Scenario) | | | Factor of 5 (Pessimistic Scenario) | | |
| | | Minimum | Mean | Maximum | Minimum | Mean | Maximum |
| Configuration A | Q = 52.9 | 90 | 100 | 110 | 225 | 250 | 275 |
| | Q = 47.4 | 79 | 88 | 97 | 198 | 220 | 242 |
| | Q = 49.0 | 86 | 96 | 106 | 216 | 240 | 264 |
| | Q = 55.6 | 97 | 108 | 119 | 243 | 270 | 297 |
| | Q = 56.4 | 97 | 108 | 119 | 243 | 270 | 297 |
| | Q = 50.1 | 90 | 100 | 110 | 225 | 250 | 275 |
| | Q = 50.7 | 86 | 96 | 106 | 216 | 240 | 264 |
| | Q = 46.8 | 79 | 88 | 97 | 198 | 220 | 242 |
| | Q = 51.8 | 86 | 96 | 106 | 216 | 240 | 264 |
| | Q = 93.3 | 68 | 76 | 84 | 171 | 190 | 209 |
| | Q = 96.6 | 72 | 80 | 88 | 180 | 200 | 220 |
| | Q = 88.2 | 61 | 68 | 75 | 153 | 170 | 187 |
| Configuration B | Q = 22.4 | 40 | 44 | 48 | 99 | 110 | 121 |
| | Q = 28.8 | 50 | 56 | 62 | 126 | 140 | 154 |
| | Q = 22.8 | 40 | 44 | 48 | 99 | 110 | 121 |
| | Q = 26.1 | 45 | 50 | 55 | 113 | 125 | 138 |
| | Q = 22.3 | 38 | 42 | 46 | 95 | 105 | 116 |
| | Q = 22.9 | 40 | 44 | 48 | 99 | 110 | 121 |
| | Q = 29.8 | 22 | 24 | 26 | 54 | 60 | 66 |
| | Q = 32.2 | 25 | 28 | 31 | 63 | 70 | 77 |

The cost is in USD. The reader is referred to Figure 5 for further details on the configuration. The reader is referred to Tables 4 and 5 for further details on the flow rates.

**Table 7.** Capital cost for each simulated design.

| Design | | Capital Cost (M$) | | | | | |
|---|---|---|---|---|---|---|---|
| | | Factor of 2 | | | Factor of 5 | | |
| | | Minimum | Mean | Maximum | Minimum | Mean | Maximum |
| Configuration A | Q = 52.9 | 104 | 114 | 124 | 244 | 269 | 294 |
| | Q = 47.4 | 91 | 102 | 112 | 215 | 239 | 261 |
| | Q = 49.0 | 100 | 110 | 121 | 233 | 259 | 283 |
| | Q = 55.6 | 111 | 122 | 134 | 260 | 289 | 315 |
| | Q = 56.4 | 111 | 122 | 134 | 261 | 289 | 315 |
| | Q = 50.1 | 103 | 114 | 125 | 244 | 269 | 295 |
| | Q = 50.7 | 99 | 110 | 121 | 233 | 259 | 283 |
| | Q = 46.8 | 92 | 102 | 112 | 216 | 239 | 260 |
| | Q = 51.8 | 99 | 110 | 121 | 235 | 259 | 284 |
| | Q = 93.3 | 81 | 90 | 99 | 189 | 209 | 228 |
| | Q = 96.6 | 85 | 94 | 104 | 199 | 219 | 239 |
| | Q = 88.2 | 74 | 82 | 90 | 170 | 189 | 208 |
| Configuration B | Q = 22.4 | 53 | 58 | 63 | 116 | 129 | 142 |
| | Q = 28.8 | 63 | 70 | 77 | 142 | 159 | 174 |
| | Q = 22.8 | 53 | 58 | 63 | 115 | 129 | 142 |
| | Q = 26.1 | 58 | 64 | 70 | 129 | 144 | 158 |
| | Q = 22.3 | 51 | 56 | 61 | 112 | 124 | 135 |
| | Q = 22.9 | 53 | 58 | 63 | 115 | 129 | 140 |
| | Q = 29.8 | 35 | 38 | 42 | 70 | 79 | 86 |
| | Q = 32.2 | 38 | 42 | 46 | 79 | 89 | 98 |

The cost is in USD. The reader is referred to Figure 5 for further details on the configuration. The reader is referred to Tables 4 and 5 for further details on the flow rates.

Sensitivity analyses indicate that, for configuration A (Figure 5), the power plant and surface facilities cost is the most influential parameter on the capital cost followed by the well cost (Figure 16). The Spearman correlation coefficients range from 0.78 to 0.91 and

0.28 to 0.54, respectively, and 0.02 and 0.13 for the stimulation costs. For configuration B (Figure 5), the power plant and surface facilities and well costs have a similar influence on the capital cost (Figure 16). The Spearman correlation coefficients vary between 0.54 and 0.70 and 0.59 and 0.73, respectively, and between 0.11 and 0.20 for the stimulation cost. Although the values displayed in Figure 15 correspond to the costs proposed by Sanyal et al. [72], the same trend is observed if the costs are 2 or 5 times higher.

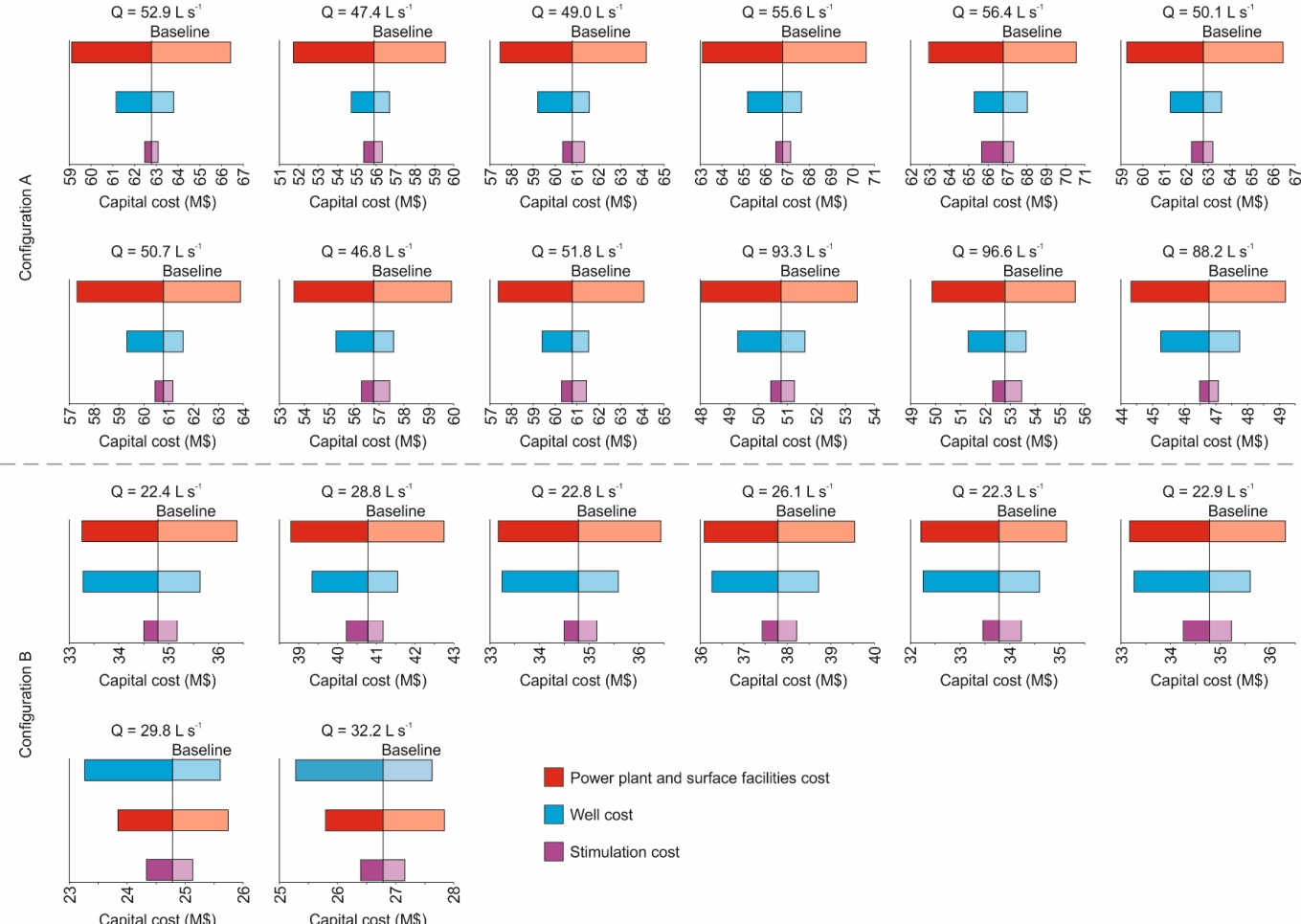

**Figure 16.** Input parameters ranked according to their influence on the capital cost. Baseline—overall simulated mean value, solid color—positive impact on the output, transparency—negative impact on the output. The cost is in USD. The reader is referred to Tables 4 and 5 for further details on the flow rates.

Considering the results previously discussed, the levelized cost of energy was evaluated considering the Sanyal et al. [72] costs as the optimistic scenario, the increase by a factor of 2 as the most likely scenario and the increase by a factor of 5 as the pessimistic scenario. The mean value for each of these hypotheses was used in this analysis. The levelized cost of energy considering configuration A (Figure 5) and the best-case scenario (Table 4) for heat production only was evaluated to range between an average minimum of 121 USD MWh$^{-1}$ and an average maximum of 626 USD MWh$^{-1}$ (Table 8). These values decreased to 120 USD MWh$^{-1}$ and 617 USD MWh$^{-1}$ if the geothermal system is assumed to work in combined heat and power mode during 10 to 15 years (Table 8). The remaining designs can only produce heat. The levelized cost of energy is found to range within an average minimum of 54 USD MWh$^{-1}$ and an average maximum of 475 USD MWh$^{-1}$ (Table 8).

**Table 8.** Levelized cost of energy for each simulated design.

| Design | | | Levelized Cost of Energy ($ MWh$^{-1}$) | | | | | |
| | | | | Heat | | | CHP | |
| | | | Optimistic | Likely | Pessimistic | Optimistic | Likely | Pessimistic |
|---|---|---|---|---|---|---|---|---|
| Configuration A | Best-case | Q = 52.9 | 136 | 247 | 584 | 134 | 244 | 576 |
| | | Q = 47.4 | 121 | 221 | 517 | 120 | 218 | 510 |
| | | Q = 49.0 | 132 | 239 | 561 | 130 | 236 | 553 |
| | | Q = 55.6 | 145 | 265 | 625 | 143 | 262 | 616 |
| | | Q = 56.4 | 145 | 265 | 626 | 143 | 262 | 617 |
| | | Q = 50.1 | 136 | 247 | 584 | 134 | 244 | 577 |
| | | Q = 50.7 | 131 | 239 | 561 | 130 | 235 | 553 |
| | | Q = 46.8 | 123 | 221 | 517 | 122 | 218 | 510 |
| | | Q = 51.8 | 132 | 239 | 563 | 130 | 235 | 555 |
| Configuration B | Base-case | Q = 93.3 | 111 | 195 | 453 | — | — | — |
| | | Q = 96.6 | 114 | 205 | 475 | — | — | — |
| | | Q = 88.2 | 102 | 178 | 410 | — | — | — |
| | | Q = 22.4 | 75 | 126 | 280 | — | — | — |
| | | Q = 28.8 | 88 | 152 | 343 | — | — | — |
| | | Q = 22.8 | 75 | 126 | 279 | — | — | — |
| | | Q = 26.1 | 82 | 139 | 312 | — | — | — |
| | | Q = 22.3 | 73 | 121 | 268 | — | — | — |
| | | Q = 22.9 | 75 | 126 | 278 | — | — | — |
| | | Q = 29.8 | 54 | 83 | 170 | — | — | — |
| | | Q = 32.2 | 58 | 91 | 192 | — | — | —- |

CHP—combined heat and power. The reader is referred to Figure 5 for further details on the configuration. The reader is referred to Tables 4 and 5 for further details on the flow rates.

Currently, the heating energy cost in Kuujjuaq has been evaluated as 0.19 USD kWh$^{-1}$ (or 190 USD MWh$^{-1}$) and electricity as 0.6 USD kWh$^{-1}$ (or 600 USD MWh$^{-1}$; [11,79]). Thus, a probabilistic study was undertaken to assess if the deep geothermal energy source, for both heat production and electricity generation, is cost-competitive compared to current oil furnaces for space heating and diesel power plants for electricity. The results for heat production reveal that configuration A for the best-case scenario have a probability of 8 to 18% of providing heating energy at a lower energy cost than the business-as-usual scenario (Figure 17a). A probability of 25 to 33% was estimated for configuration A for the base-case scenario (Figure 17b), while configuration B reveals probabilities of 49 to 91% (Figure 17c). Per contra, the probability of combined heat and power to provide electricity at a lower cost than the business-as-usual scenario is higher than 99% (Figure 17d). This analysis was undertaken by combining the results obtained for the optimistic, most likely and pessimistic scenarios. Considering the most likely scenario only, the engineered geothermal energy system has 13 to 100% probability of providing heating energy at a lower cost than the oil furnaces and 100% probability of providing electricity at a lower cost than the diesel power plants.

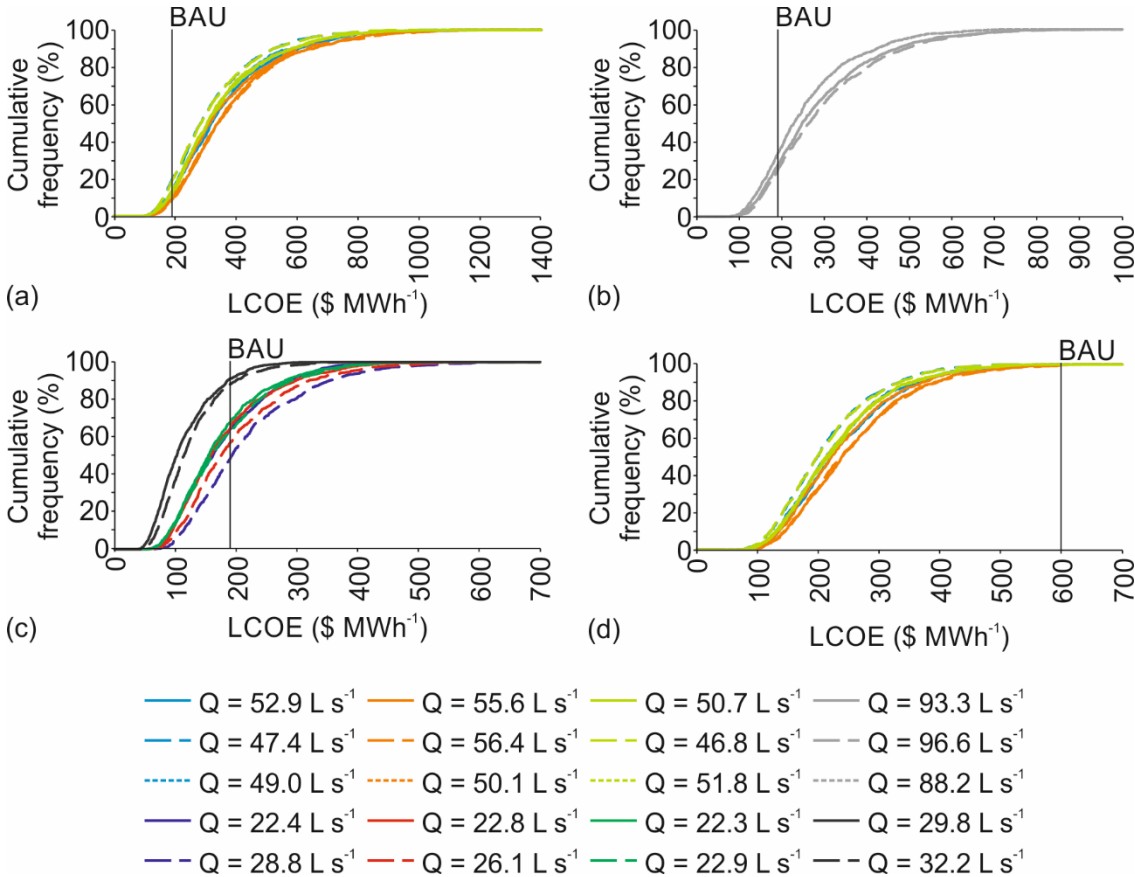

**Figure 17.** Levelized cost of energy and probability of providing energy at lower cost than the business-as-usual (BAU) scenario: (**a**) configuration A and best-case scenario—heat only; (**b**) configuration A and base-case scenario—heat only; (**c**) configuration B—heat only; (**d**) configuration A and best-case scenario—combined heat and power. The cost is in USD. The reader is referred to Tables 4 and 5 for further details on the flow rates.

A detailed analysis was undertaken to assess how the probability of providing affordable heating energy with an engineered geothermal energy system can be increased. The results revealed that the energy consumption and capital cost play the major roles, with Spearman coefficients of, on average, −0.70 and 0.60, respectively. The operation and maintenance costs, per contra, reveal a weak correlation (on average 0.02) with the levelized cost of energy. This trend is observed for all of the simulated scenarios (Figure 18). In fact, for the best- and base-case scenarios assuming configuration A, if the energy consumption is near or greater than its maximum value and the capital cost near or below its minimum value, then the geothermal systems simulated will have a greater probability of providing heating energy at a lower cost than the business-as-usual scenario (Figure 19). However, for configuration B at the best-case scenario, the energy consumption only needs to be greater than its 30th to 50th percentile and the capital cost below, its 50th percentile (Figure 20). For the base-case scenario, in turn, the energy consumption can be near its minimum value and the capital cost near its maximum value (Figure 20).

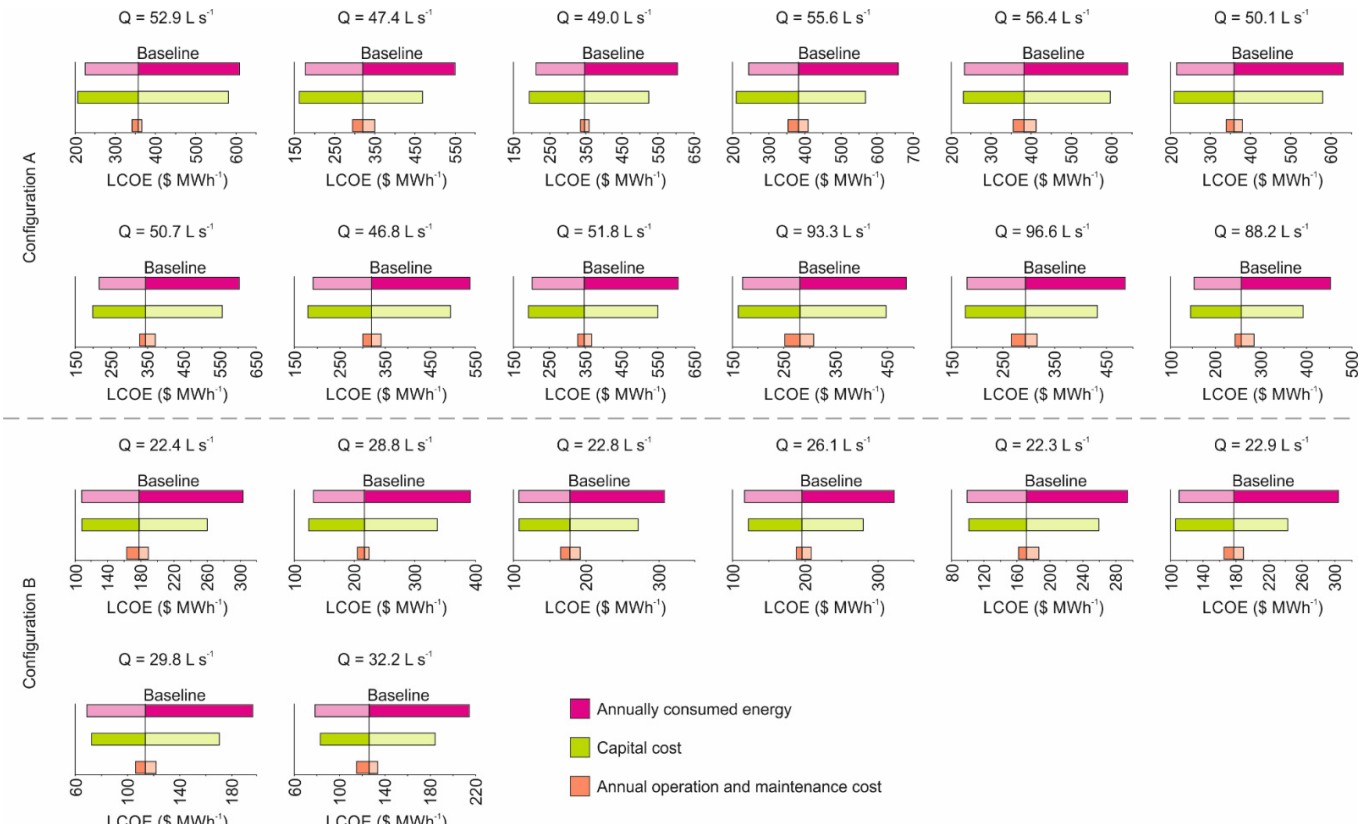

**Figure 18.** Input parameters ranked according to their influence on the levelized cost of energy. Baseline—overall simulated mean value, solid color—positive impact on the output, transparency—negative impact on the output. The cost is in USD. The reader is referred to Tables 4 and 5 for further details on the flow rates.

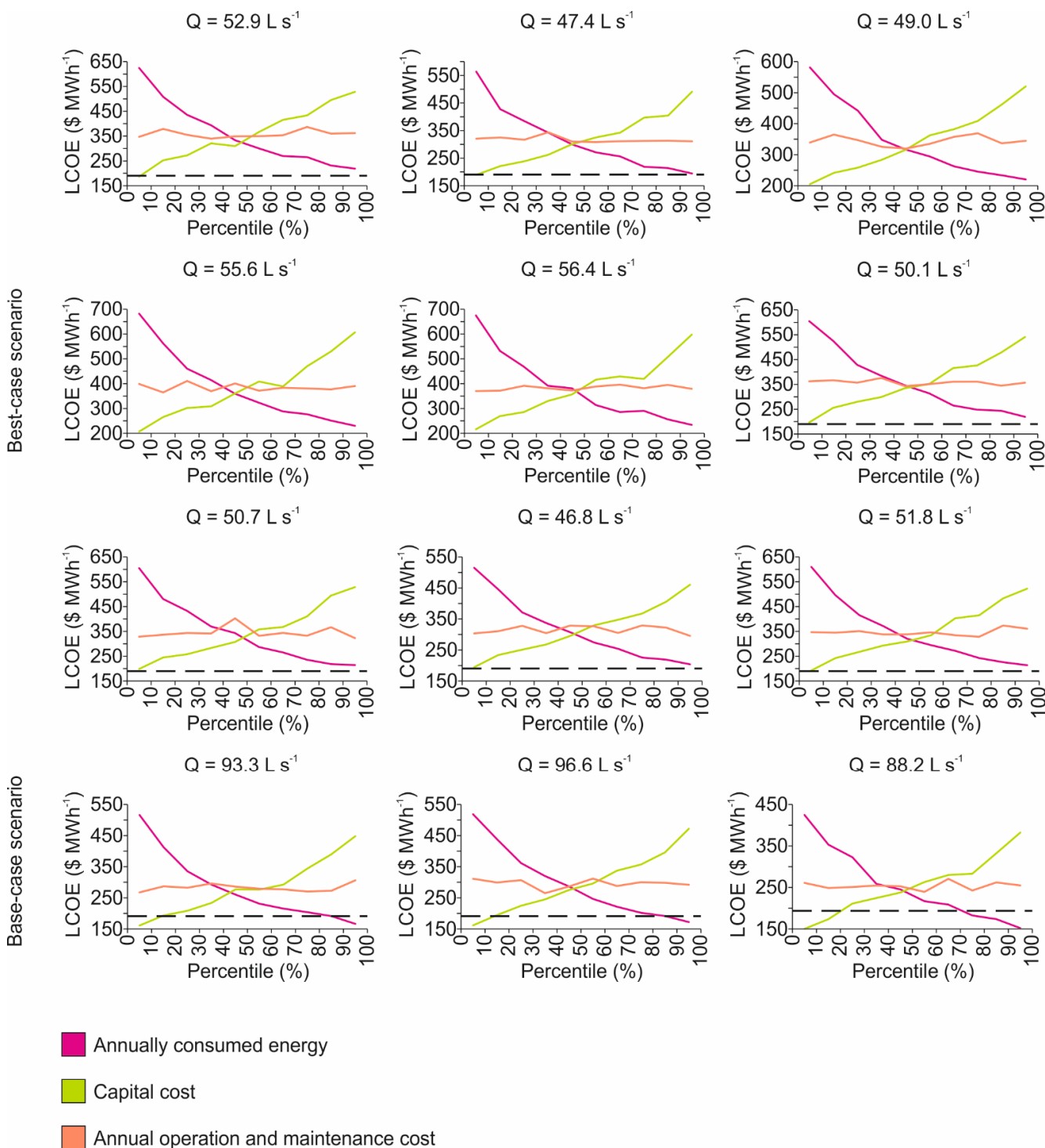

**Figure 19.** Levelized cost of energy as a function of the uncertain parameters' percentile for configuration A. Dashed line—heating energy cost with oil furnaces (see text for further details). The reader is referred to Tables 4 and 5 for further details on the flow rates.

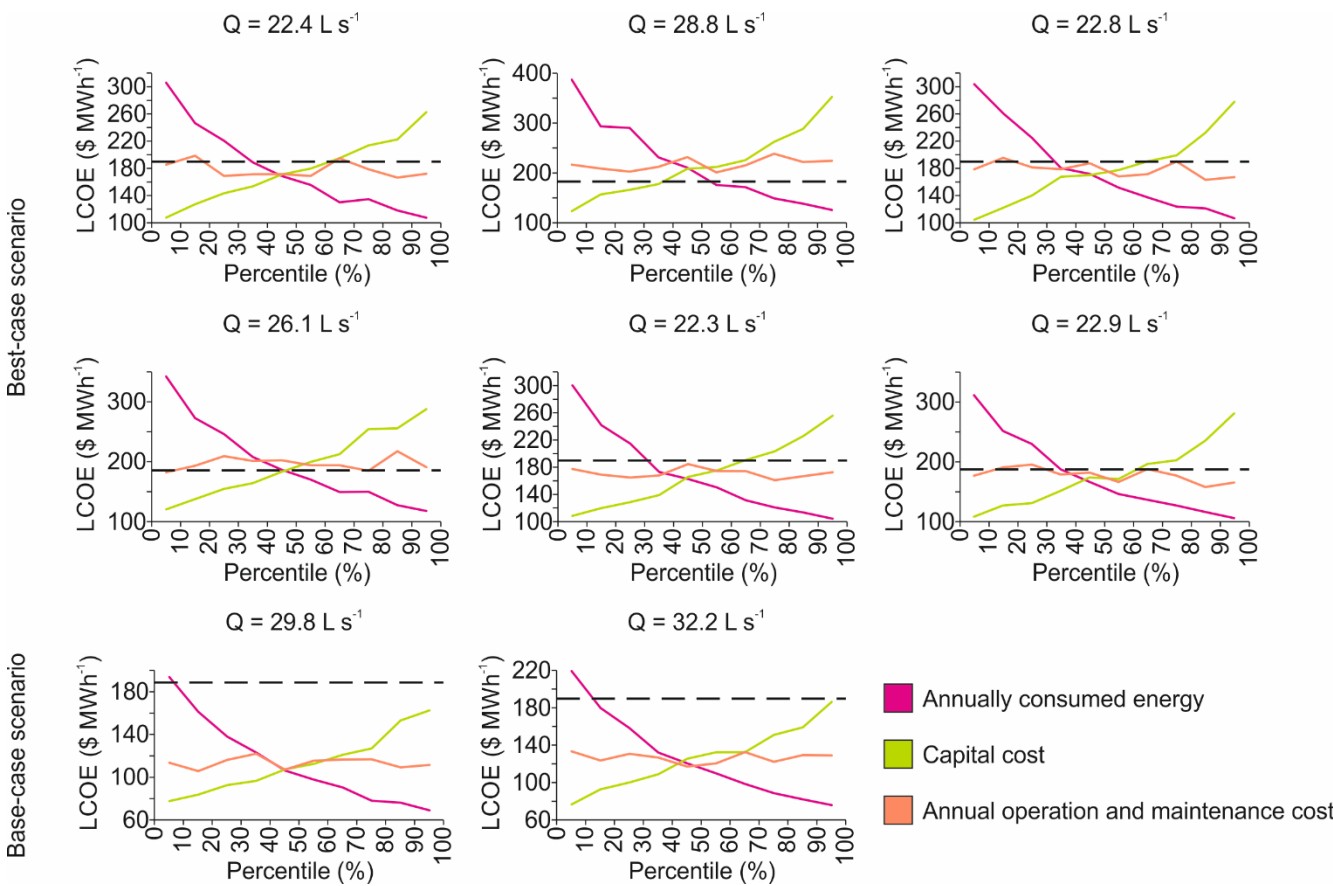

**Figure 20.** Levelized cost of energy as a function of the uncertain parameters' percentile for configuration B. Dashed line—heating energy cost with oil furnaces (see text for further details). The reader is referred to Tables 4 and 5 for further details on the flow rates.

## 6. Discussion

The current energetic framework of the 239 Canadian off-grid communities relying solely on diesel entails: (1) high costs to buy, transport and store diesel, (2) carbon emissions that contribute to climate change, (3) potential spills and leakages that damage the local environment and (4) low energy security that constrains community development (e.g., [80]); thus, opposite to the three main axes that constitute the concept of sustainable energy markets: economic affordability, environmental compatibility and energy security [69,81]. A sustainable energy market intends to maintain and improve living standards at an affordable cost by replacing the consumption of environmentally harmful sources of energy by environmentally friendly alternatives. Therefore, introducing renewable off-grid technologies into the communities can break this energy poverty cycle that has held back their socio-economic progress. Deep geothermal energy can be a viable alternative to fossil fuels in remote northern communities, if the following criteria are met [82]:

- It is sufficiently abundant to meet a significant percentage of the market demand;
- It can be obtained at a cost competitive with existing energy sources.

A previous deep geothermal energy source assessment suggested that the thermal energy stored beneath the community of Kuujjuaq at depths greater than 4 km was capable of fulfilling an estimated community heating energy demand [28]. Thus, the current study was carried out with the goal of designing an engineered geothermal energy system to harvest this energy and evaluating its economic potential compared to the current use of fossil fuels. However, the results from this study are subjected to high levels of uncertainty due to the current poor knowledge of both the geological structures and thermo-hydro-mechanical properties of rock at depth in the area. Nevertheless, this work is a contribution

to assess if deep geothermal energy sources can supplant or displace reliance on fossil fuels. Moreover, although this work was undertaken in the community of Kuujjuaq, the methodological approach followed can be extended to other remote northern settlements facing important energy issues and with similar geothermal exploration data gaps.

The first research questions of this study were: Will the hydraulic stimulation technique applied in crystalline basement rocks develop a well-connected flowing system in Kuujjuaq? How can this be done? What further local geological and thermo-hydro-mechanical data is required for more accurate predictions? A shear-dilation-based model (FRACSIM3D; [52]), updated for new joint constitutive laws, was used in this work to help on the design of an engineered geothermal energy system for each uncertain geological (fracture network) and thermo-hydro-mechanical (properties of the medium) scenario. Although McClure and Horne [83] propose different hydraulic stimulation mechanisms (pure opening mode, pure shear stimulation, primary fracturing with shear stimulation leakoff and mixed-mechanism stimulation), shear displacement is still the most widely accepted mechanism to explain permanently enhanced transmissivity of natural fracture networks (e.g., [44,84,85]). Jacking (i.e., fracture-normal dilation) can also occur near the open-hole section, but its effect declines over time with the depressurization (e.g., [84,85]), but may increase over time with thermoelastic effects following cooling. Nevertheless, field observations at the Fenton Hill test site have supported mixed-mechanism stimulation, involving the propagation of hydraulic splay fractures due to stress changes induced as fractures opened and failed to shear [86]. Thus, the present state of knowledge highlights the importance of continued field tests to better characterize the subsurface and understand stimulation mechanisms.

Studies have shown that considering the effect of thermoelasticity tends to improve the flow rates, decrease water loss and reduce the hydraulic impedance (e.g., [87]). In this study, although a thermoelasticity module has already been developed for an earlier version of FRACSIM3D [87], the authors focused only on the hydro-mechanical and thermo-hydro coupling processes. Nevertheless, future improvements could include this module and assess the effects of considering only poroelasticity and considering both thermo and poroelasticity effects. The simulations reveal that the development of hydraulically stimulated geothermal reservoirs is favored in a high temperature, hydraulically conductive and mechanically weak (in terms of low magnitude of the principal stresses and low resistance of fractures to deformation and opening) subsurface, i.e., the best-case scenario assumed in this study. If the reservoir temperature and subsurface hydraulic conductivity are decreased and the magnitude of the principal stresses and the fractures resistance to deformation and opening is increased, then developing engineered geothermal energy systems is more difficult. In fact, the base-case scenario assumed in this study only generated reservoirs of potential interest assuming fracture longer than those sampled in the field. Moreover, if the worst-case scenario is the prevailing scenario in the study area, then engineered geothermal energy systems are not a viable alternative off-grid technology to offset fossil fuels' consumption in the community of Kuujjuaq. Nevertheless, the work conducted suggests additional geothermal exploration to improve these predictions is warranted. Exploratory boreholes deeper than 300 m are necessary to carry out more accurate estimations of the subsurface temperature, stress field regime and fracture network. Hydraulic stimulation field tests are also required to obtain more accurate information on the hydro-mechanical behavior of the subsurface. Another important aspect to accurately assess is the characterization of the fault hydraulic properties that may play a major role for the development of engineered geothermal energy systems in Kuujjuaq. Although numerical models are fundamental to support the design and operational planning of engineered geothermal energy systems, reservoir simulation is a complex endeavor that requires careful calibration [88]. Hydraulic stimulation is accompanied by induced seismicity and the cloud of events is an important monitoring tool to understand the direction in which the reservoir can develop (e.g., [34,63,89]). Moreover, fracture seismic observations recorded before, during and after hydraulic stimulation enable the building of time-lapse

images mapping the fluid flow in the rocks, and thus estimate the reservoir connectivity changes over time (e.g., [90]). The seismic method used to analyze fracture behavior during stimulation is based on the recording and analyzing of passive seismic data. Drilling the wells without previous knowledge of the reservoir growth direction, jointing pattern and in situ stress field can severely compromise a stimulated geothermal project (e.g., [34,89]). A lesson learned from Rosemanowes's stimulated geothermal project is that the wells should be drilled with their azimuths parallel to the minimum in situ horizontal stress direction (e.g., [43]). In the current work, the wells were assumed to be vertical, but further simulations can be envisioned by changing their dip, dip directions and relative azimuths. However, the uncertainty associated with the direction of the stress field must be resolved at an early stage. Microseismic monitoring can also be used to infer the fracture shear displacement (e.g., [63]). Additionally, an artificial neural network–genetic algorithm-based displacement back analysis has been proposed by Zhang et al. [91] to estimate fracture stiffness, in situ stresses and elastic parameters from borehole displacements during drilling. Laboratory tests are also an important component to enhance the understanding of joint behavior (e.g., [92–99]). However, care must be taken due to scaling effects. Nevertheless, the laboratory tests would provide first-order calibrations for the fractures opening parameters. Laboratory tests were not in the scope of this study but may be beneficial to refine the range of applicable input parameters. Moreover, further simulations can be carried out by changing the configuration from a doublet to a triplet, for instance. Two injectors and one producer may help to develop hydraulically stimulated geothermal reservoirs for the base-case scenario with shorter fractures. This can, however, impact the capital cost attributable to wells.

Chemical mineral dissolution by alkaline or acidic additives was used at Soultz to clean joints, faults and pore volume and, thus, helping to enhance the natural permeability of these structures [44,100]. The application of chemical stimulation may be an additional option for further studies in Kuujjuaq if future geothermal exploration reveals the presence of fractures infilling material that can react to this procedure. If this is observed, then a 3D water/rock chemical interaction module has been developed for FRACSIM3D [101] that can be used to simulate both hydraulic and chemical stimulations and evaluate the system performance response. Moreover, laboratory experiments with lithologies analogous to the potential reservoir rock in Kuujjuaq can provide further insights as to whether chemical stimulation may be effective (e.g., [102]). Thermal stimulation can also be applied to increase the near-wellbore productivity (e.g., [103,104]).

The numerical simulations carried out considered water as the working fluid. Further improvements can be carried out to assess the change in performance if supercritical $CO_2$ is used instead. This concept was proposed by Brown [105] and the advantages of operating engineered geothermal energy systems with supercritical $CO_2$ are twofold: (1) $CO_2$ has certain thermophysical and chemical properties that make it an attractive heat transfer medium and (2) such systems can promote geological storage of $CO_2$ as an ancillary benefit. Numerical simulations have been carried out by several authors, and their results suggest a significant performance improvement compared to water [106–108]. Pruess's [106] numerical simulations reveal thermal extraction rates approximately 50% larger for $CO_2$ compared to water. This suggests that flow rates 50% lower than the ones currently estimated for Kuujjuaq may be enough to extract the same amount of energy. In 2012, the 14 diesel power plants in Nunavik contributed with 65,000 tonnes/year of $CO_2$ equivalent [109]. This equals to a flow rate of approximately 2.1 L s$^{-1}$. Assuming that the required flow rate to meet Kuujjuaq's minimum heating demand threshold is now 7 L s$^{-1}$ instead of the 14 L s$^{-1}$ estimated using water as the working fluid, then 3.5 years of capture and storage of $CO_2$ are needed to reach that flow. This suggests that although $CO_2$ can be captured from the thermal plants, used in engineered geothermal energy systems (e.g., [110]) and its thermal performance was proven superior to water (e.g., [106]), this solution is not technically or economically viable for northern communities. For example, the amount of $CO_2$ produced by a single diesel power plant is insufficient

to meet Kuujjuaq's demand and shipping additional $CO_2$ from other communities and southern areas can increase the costs of the geothermal project and may bring additional environmental impacts. A review of large-scale $CO_2$ shipping has been undertaken by Baroudi et al. [111] and the transport costs range between USD 10 and USD 167 per tonne $CO_2$, depending on the travel distance and transport capacity. An alternative working fluid, supercritical $N_2O$, was proposed by Olasolo et al. [112] due to its thermodynamics properties but may still not be a viable solution for remote northern communities.

The second research question was: Are the deep geothermal energy sources harvested by engineered geothermal energy systems in Kuujjuaq cost-competitive compared to fossil fuels? The levelized cost of energy was estimated based on literature values for the US proposed by Sanyal et al. [72] and increased by factors of 2 and 5 to be more in the range expected in remote northern regions. Empirical well cost models were also used (e.g., [71] and references therein). Two 4-km-deep wells were inferred to cost from USD 9.8 to 12.1 M. However, Minnick et al. [25] refer that in Nunavut (northern Canada) a full-size 4-km production well can cost approximately USD 12 M. This corresponds to twice the estimated price in this work since Minnick et al.'s [25] estimate includes expenses associated to Nunavut's challenging environment. Such a difference leads to an increase of 14 to 31% of the capital cost, which in turn increases the levelized cost of energy. The levelized cost of energy, considering the literature well cost models and assuming heat production only, ranges between 83 USD MWh$^{-1}$ and 265 USD MWh$^{-1}$ for the likely scenario. Doubling the well costs to be in line with anticipated Nunavik costs leads to values varying between 108 USD MWh$^{-1}$ and 345 USD MWh$^{-1}$.

Nevertheless, the first-order evaluation carried out in this study suggests that engineered geothermal energy systems may have commercial interest. The probability of geothermal heat production being less expensive than the business-as-usual scenario (oil furnaces) ranges from a minimum of 8% to a maximum of 91%. This probability can be increased if the energy consumption of the community is near or above its maximum estimated value and if the capital cost is decreased to its minimum value.

Furthermore, if the best-case scenario prevails in the study area, then combined heat and power may be a viable option to not only increase the sustainability of the geothermal system, but also to decrease the levelized cost of energy and, thus, increase the economic potential. In fact, combined heat and power have more than 99% of probability to provide electricity at a lower cost than the current diesel power plants. It is important to highlight that the geothermal system was assumed to be working 24 h per 7 days during the 30 years of project lifetime. The most likely levelized cost of energy, for heat production only, in this study, was found to range between 83 USD MWh$^{-1}$ and 265 USD MWh$^{-1}$. For combined heat and power, the most likely levelized cost of energy ranges between 218 USD MWh$^{-1}$ and 262 USD MWh$^{-1}$. These values are within the range referred by Tester et al. [35] and Augustine [113], 100 to 1000 $USD MWh$^{-1}$ and 140 to 310 USD MWh$^{-1}$, respectively. However, Tester et al. [35] also mentioned that with mature and cheaper technology, the levelized cost of energy can reach values as low as 36 to 92 USD MWh$^{-1}$. Note that the levelized cost of energy described by Tester et al. [35] and Augustine [113] are for electricity generation only and not heat production or combined heat and power as studied in Kuujjuaq.

A previous economic potential assessment of geothermal energy sources in northern Canada undertaken by Majorowicz and Grasby [23] suggests a total cost of the project of USD 26 M. These estimations were carried out for a depth of 3 to 5 km, doublet spaced 550–700 m, reservoir temperature of 120 °C and flow rate of 30 L s$^{-1}$. The remaining estimations presented by these authors were done for electricity generation only, thus difficult to compare with the results obtained in the current study. Nevertheless, these authors evaluated a cost ranging from 0.50 to 0.84 USD kWh$^{-1}$. Additionally, Richard [114] carried out an assessment of the viability of engineered geothermal energy systems for electricity generation in Québec. The base-case estimated production costs were found to vary between 1.25 USD kWh$^{-1}$ at 4 km and for a reservoir temperature of 100 °C and

0.18 USD kWh$^{-1}$ at 10 km depth and 250 °C using a reduced drilling cost. In this work, an engineered geothermal energy system in Kuujjuaq for direct use applications indicates a most likely cost of 0.08 to 0.27 USD kWh$^{-1}$, while combined heat and power is 0.22 to 0.26 USD kWh$^{-1}$. Thus, deep geothermal energy sources harvested by engineered geothermal energy systems may be cost-competitive compared to fossil fuels, and further gathering of information is worthwhile to improve the quality of these predictions.

### 6.1. Comparison with Other Deep Geothermal Energy Projects

Engineered geothermal energy systems, or enhanced geothermal systems (EGS), are always site-specific in terms of both the subsurface environment and the local energy demand and price sensitivity. Thus, the comparison of the results obtained in this study with other deep geothermal energy projects developed in different geologic–economic contexts is challenging. The thermo-hydro-mechanical characteristics of the subsurface, which highly influence the design and development of EGS, vary from place to place. To date, the technological and economic feasibility of EGS in the Canadian Shield has not yet been fully assessed. Due to a lack of field experiments, history matching to calibrate the numerical simulations and restricting the range of assumptions made in this study is not possible. Nevertheless, in order to design an EGS for Kuujjuaq, the approach was strongly influenced by the results obtained and the lessons learned from the hydraulically stimulated geothermal projects of Fenton Hill, Rosemanowes, Soultz and Rittershoffen [34,41–44], many of which are incorporated into the design of the FRACSIM3D model. For example, the acceptable targets chosen for water loss, thermal impedance and thermal drawdown were based on targets adopted for those geothermal projects (e.g., [34,41–44]) even though the economics at Kuujjuaq differ substantially. The flow rates applied to the numerical models also considered the values targeted and observed in the commercial projects of Soultz and Rittershoffen [68]. The effective stimulation pressure required to induce shear slip in the pre-existing natural fractures is also found to be comparable to the values found at those four geothermal sites [37]. The potential heat and power output from an EGS developed in Kuujjuaq is also similar to the power generated and heat produced at Soultz and Rittershoffen [68]. The Soultz-sous-Forêts geothermal power plant is producing electricity using an organic Rankine cycle and has an installed gross capacity of 1.7 MW$_e$. The results of this study suggest a potential electricity production of 2.9 to 3.5 MW$_e$ provided that the "best case" in situ temperatures are found. The Rittershoffen geothermal plant is providing superheated water for industrial needs and has an installed capacity of 24 MW$_{th}$. The results in Kuujjuaq suggest a potential heat production between 5.7 and 29 MW$_{th}$. Finally, the capital cost and levelized cost of energy are also within values published for northern Canada and southern Québec [23,114].

A geothermal cascade system can be an option for the deep geothermal energy source in Kuujjuaq as it has been proposed in southern Poland [115]. In southern Poland, a deep geothermal well with a geothermal water temperature of 82 °C and a maximum flow rate of 51.22 kg s$^{-1}$ has been considered to assess the potential for electricity generation. The thermodynamic calculations carried out by Kaczmarczyk et al. [115] indicate that the gross capacity in the most optimistic variant will not exceed 250 kW for the organic Rankine cycle and 440 kW for the Kalina cycle, and that the gross electricity generation will not exceed 1.9 GWh/year for the organic Rankine cycle and 3.5 GWh/year for the Kalina cycle. If the same order of values is found in Kuujjuaq, electricity generation from geothermal waters of 82 °C and flow rates of 51 kg s$^{-1}$ does not seem to be a viable option to replace diesel since the annual average electricity demand in the community is almost 19 GWh. Nevertheless, the use of geothermal energy, regarding the applications, has several positive aspects, both environmental and social, that can significantly improve the living conditions at Kuujjuaq and, indeed, in any region as highlighted by, for instance, Operacz and Chowaniec [116] and Sowizdzal et al. [117].

*6.2. Comparison with Other Renewable Energy Alternatives*

Another geothermal solution to reduce diesel consumption in remote northern regions is ground-coupled heat pumps [15] and borehole thermal energy storage [118]. Within the components of a shallow geothermal system exploited by ground-source heat pumps, the ground heat exchanger is a key component, playing a major role in achieving a high coefficient of performance [119]. Important parameters for the ground heat exchangers include the geometry, the pipe material, the backfill material, the working fluid, the depth of the ground heat exchanger, the heat transfer coefficient, the outlet temperature, the thermal resistance and the pressure drop [119–121]. Although ground-source heat pumps can be viable to supply heat to residential dwellings [15], the extracted energy with such shallow systems (100 to 300 m) is generally insufficient to fulfill the heating demand of a single residential dwelling, relying on an energy input to cover the remaining load.

Biomass, wind, solar, biofuels and hydro have also been studied as alternatives to offset diesel consumption in remote Canadian communities [6,7,10,12,14,17]. Thompson and Duggirala [7] and Yan et al. [14] indicate biomass as the most favorable and competitive renewable energy technology compared to natural gas, gasification of domestic waste, wind and solar. Furthermore, Stephen et al. [10] carried out a study to determine the techno-economic feasibility of biomass utilization for space heating and concluded that biomass has the potential to reduce heat costs, reduce the cost of electricity subsidizations for electrical utilities, reduce greenhouse gas emissions and increase energy independence. However, biomass resources need to be transported and stored similarly to diesel, which is disadvantageous compared with the development of a local source of energy. A fully hybrid wind-solar-battery-diesel system was selected by a multi-objective genetic algorithm as a viable solution for northern communities [17]. This system suggests a reduction of 50% of the levelized cost of energy compared to a diesel-only scenario and it may help to displace 675 MWh/year of energy from diesel. However, wind and solar remain weather-dependent and, therefore, their supply of energy is intermittent [122]. McFarlan [12] conducted a techno-economic assessment of replacing diesel for electricity generation with clean biofuels (methanol and dimethyl ether). The results revealed an increase of the cost compared to the diesel scenario. However, McFarlan [12] argued that although clean biofuels are more expensive than diesel there is potential socio-economic benefits from switching to these energy sources. Biofuels are still in a nascent stage and more research and development are needed. Micro-hydropower systems in off-grid communities have been identified as a favorable solution to displace the use of diesel and reduce greenhouse gas emissions [6]. Most of these systems are run-of-river without a need for a dam or reservoir. The best geographical locations would be steep rivers, streams, creeks or springs flowing year-round [6]. Although water resources may not be a current issue throughout Canada, climate change may impact existing and proposed hydropower projects, especially in the boreal, subarctic, arctic unique and complex environments [123].

## 7. Conclusions

Geothermal energy off-grid technologies are a potential solution for improving the energetic framework of the 239 Canadian remote northern communities that rely solely on diesel for electricity and space heating. Although subject to high uncertainty, the results of this work suggest that engineered geothermal energy systems are technically and economically viable in the community of Kuujjuaq (Nunavik, Canada) to supplant the reliance on fossil fuels. A "what-if" approach was followed in this study to deal with the poor subsurface knowledge. The engineered geothermal energy systems designed for each highly uncertain scenario needed to provide enough thermal energy for direct use applications during 30 years of operation and to be within certain defined performance parameters limits. These were: water loss lower than 20%, reservoir flow impedance lower than 1 MPa L$^{-1}$ s$^{-1}$ and thermal drawdown lower than 1 °C/year. The numerical simulations revealed that developing hydraulically stimulated geothermal reservoirs is favored in a high temperature, hydraulically conductive and mechanically weak subsurface

(in terms of low magnitude of the principal stresses, high differential stress and low resistance of fractures to slip and opening). Decreasing the reservoir temperature and hydraulic conductivity and increasing the magnitude of the principal stresses and the fractures resistance to slip and opening decreases the performance of the system. Moreover, where an NW-SE striking fault is present, placing the wells relative positions parallel to the inferred direction of the maximum principal stress (NE-SW) revealed better performance than if the wells are located parallel to the fault plane. This may also be caused by an overlap of the stimulation volumes. In fact, the simulation results revealed to be relatively insensitive to the assumed initial fault apertures, suggesting that the fault itself plays little role in fluid flow circulation. However, if the fault segment runs quasi-parallel to the maximum principal stress (and so being capable of further stimulations) may make a significant difference to the results. Furthermore, longer fractures tend to improve the performance of the system. Smaller fractures have higher shear stiffness and tend to slip less, making fluid circulation more difficult. Additionally, the best-case scenario suggests that combined heat and power is possible during the first 10 to 15 years of the geothermal system operation if the in situ temperature assumptions are met.

A first-order evaluation of the capital cost and levelized cost of energy was carried out using the Monte Carlo method. The results for capital cost is in the range USD 25 to 67 M$ for the optimistic scenario, USD 38 to 122 M for the likely scenario and USD 79 to 289 M for the pessimistic scenario. The global sensitivity analysis based on the correlation coefficient revealed that the power plant and surface facilities cost is the most influential parameter on the capital cost, followed by the well cost and stimulation cost. The levelized cost of energy, assuming heat production only, was estimated to range within 54 and 145 USD MWh$^{-1}$ for the optimistic scenario, between 83 and 265 USD MWh$^{-1}$ for the likely scenario and between 170 to 626 USD MWh$^{-1}$ for the pessimistic scenario. Combined heat and power revealed a levelized cost of energy varying between 120 and 143 USD MWh$^{-1}$ for the optimistic scenario, 218 and 262 USD MWh$^{-1}$ for the likely scenario and 510 and 617 USD MWh$^{-1}$ for the pessimistic scenario. The probabilistic analysis carried out indicates that engineered geothermal energy systems have an 8 to 91% probability of providing heating energy at a lower cost than the current oil furnaces and more than a 99% chance of providing electricity at a lower cost than the diesel power plants currently in place. Given that geothermal energy is a local source available for remote community, further geothermal exploration is recommended and indispensable to decrease the existing uncertainties and support decisions to develop this energy alternative. Hence, helping remote northern communities moving towards a greener and sustainable energetic future.

The work described in this study highlights several uncertain geological and thermo-hydro-mechanical parameters that need further gathering of information to obtain more accurate estimates of the techno-economic potential of EGS in Kuujjuaq. Exploratory boreholes deeper than 1 km are required for accurate heat flux and subsurface temperature assessments (e.g., [34,42–44,124,125]). These boreholes are also needed to carry out hydraulic tests (e.g., [34,42–44,124]) and stress measurements (e.g., [34,42–44,124]). Borehole televiewer can also be useful to have a better characterization of the underground fracture network (e.g., [34,42–44,124]). The effect of temperature (or thermal lift) on the hydraulic properties (e.g., [126]) could be additionally evaluated. Thus, the next step that is justified by this research is the drilling of exploration boreholes to obtain more accurate data for the numerical simulations that can be improved with history matching.

**Author Contributions:** Conceptualization, M.M.M., J.R., J.W.-R. and C.D.; methodology, M.M.M. and J.W.-R.; software, J.W.-R.; validation, M.M.M.; formal analysis, M.M.M.; investigation, M.M.M.; resources, M.M.M., J.R., J.W.-R. and C.D.; data curation, M.M.M.; writing—original draft preparation, M.M.M., J.R., J.W.-R. and C.D.; writing—review and editing, M.M.M., J.R., J.W.-R. and C.D.; visualization, M.M.M.; supervision, J.R. and C.D.; project administration, J.R.; funding acquisition, J.R. All authors have read and agreed to the published version of the manuscript.

**Funding:** This study was funded by the Institut nordique du Québec (INQ) through the Chaire de recherche sur le potentiel géothermique du Nord awarded to Jasmin Raymond. The Centre d'études nordiques (CEN), supported by the Fonds de recherche du Québec—nature et technologies (FRQNT), and the Observatoire Homme Milieu Nunavik (OHMI) are further acknowledged for helping with field campaigns cost and logistics.

**Institutional Review Board Statement:** Not applicable.

**Informed Consent Statement:** Not applicable.

**Data Availability Statement:** All the relevant dataset are presented in this study.

**Acknowledgments:** The authors would like to acknowledge Félix-Antoine Comeau, Inès Kanzari, Jean-François Dutil, Sérgio Seco and Stefan Premont for the support during the analyses of the thermophysical properties. Acknowledgments are extended to Cynthia Brind'Amour-Cote for the support during the field campaign. A special thanks is given to Fiona Chapman for the English language review. The authors are also grateful to the three anonymous reviewers whose comments and suggestions helped to make the manuscript clearer.

**Conflicts of Interest:** The authors declare no conflict of interest. The funders had no role in the design of the study; in the collection, analyses or interpretation of data; in the writing of the manuscript, or in the decision to publish the results.

## Notation

| Symbol | Definition | Unit |
|---|---|---|
| $A$ | Area | $m^2$ |
| $C$ | Cost | $ |
| $e$ | Thermal energy | W; MWh |
| $F$ | Factor | — |
| $H$ | Hydraulic impedance | $MPa\,L^{-1}\,s^{-1}$ |
| $i$ | Imputed interest rate | % |
| $I$ | Total capital investment | $ |
| $I^*$ | Injection well | — |
| $l$ | Length | m |
| $LCOE$ | Levelized cost of energy | $\,MWh^{-1} |
| $N$ | Number | — |
| $O$ | Annual operation and maintenance cost | $¢\,kWh^{-1}$ |
| $P$ | Pressure | Pa |
| $\nabla P$ | Pressure gradient | $Pa\,m^{-1}$ |
| $Q$ | Flow rate | $m^3\,s^{-1}$ |
| $q$ | Heat flux | $mW\,m^{-2}$ |
| $R^*$ | Recovery well | — |
| $T$ | Temperature | °C; K |
| $t$ | Time | s; year |
| $TDS$ | Total dissolved solids | $kg\,L^{-1}$ |
| $U$ | Shear displacement | m |
| $u$ | Darcy velocity | $m\,s^{-1}$ |
| $V$ | Volume | $m^3$; $km^3$ |
| $W$ | Water | % |
| $w$ | Aperture | m |
| $z$ | Depth | m |
| Greek letters | | |
| $\lambda$ | Thermal conductivity | $W\,m^{-1}\,K^{-1}$ |
| $\rho c$ | Volumetric heat capacity | $J\,m^{-3}\,K^{-1}$ |
| $\sigma$ | Principal stress | Pa |
| $\sigma'_n$ | Effective normal stress | Pa |
| $'_{n\_ref}$ | Reference stress for 90% closure | Pa |
| $\tau$ | Shear stress | Pa |
| $\phi$ | Angle | ° |
| $\omega$ | Dynamic viscosity | $kg\,m^{-1}\,s^{-1}$ |

| Subscript | |
|---|---|
| 0 | Initial |
| circ | Circulation |
| H | Maximum horizontal principal stress |
| h | Minimum horizontal principal stress |
| rec | Recovered |
| s | Scaling |
| stim | Stimulation |
| th | Thermal |
| V | Vertical principal stress |
| Abbreviation | |
| HSD | Hydrothermal spallation drilling |

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
