# Peer review of "Are Engineered Geothermal Energy Systems a Viable Solution for Arctic Off-Grid Communities? A Techno-Economic Study"

_water, doi:10.3390/w13243526_

Round 1
Reviewer 1 Report
Very interesting paper. The paper transmits all the work behind the paper. A very high scientific approach to a challenging issue.
Author Response
Dear reviewer,
We would like to thank your contribution. Your comments were carefully examined and incorporated in the new version of the manuscript. In the following lines, we have addressed each specific comment and question, indicating how the modifications were incorporated in the manuscript.
R: Extensive editing of English language and style
The English language and style were revised by native-English colleagues, and we believe that the language and style have been significantly improved. The changes made are listed below:
Line 13 – “circulated” to “circulating fluids in”
Line 15 – “(…) large uncertainty still exists due to the current poor knowledge of both relevant geology and thermo-hydro-mechanical data.” to “(…) poor knowledge of relevant geology and thermo-hydro-mechanical data introduces significant uncertainty in numerical simulations.”
Line 16 – “Nevertheless” to “Here”
Lines 17 – deleted “provide a range of possibilities and (…) to (…)”.
Line 19 – “that provides” to “Each possibility meets (…)”
Line 26, 73, 840, 1107 – “bounded by” to “subject to”.
Line 32 – “clean” to “an environmentally benign”
Line 37 – “trend” to “predominance”
Line 38 – “watchwords” to “buzzwords”
Line 61 – “detailed yet broadly based prediction” to “comprehensive estimation”.
Line 63 – “in this physiographic region” to “the Canadian Shield”
Line 76 – “and neither was” to “nor were they”
Line 81, 82 – “lower than” to “below”
Line 84 – “such systems” to “engineered geothermal energy systems”
Line 276 – “The most significant aspect of this new sliding/opening law is that the rate of opening with displacement can be made to reduce (…)” to “The rate of opening with displacement associated with this new sliding/opening law is (…)”
Line 280 – “becomes large enough” to “exceeds threshold”
Line 332 – “(…) is now allowing (…) of microsesimic events of appropriate magnitude and in appropriate numbers.” to “(…) allows (…) of microsesimic events of appropriate magnitude and numbers (…)”
Line 357 – “A (…) to compensate for the flow overestimation typically given by the parallel plate assumption when applied to rough natural fractures. Experiments have suggested that the mechanical aperture of rough fracture overestimates the flow capacity to some extent.” To “Experiments have suggested that the mechanical aperture of rough fracture overestimates the flow capacity due to the parallel plate assumption. Therefore, a (…)”
Line 374 – “is cooled own” to “cools”
Line 388 – “allowed” to “are able”
Line 391 – “never gets close” to “happens away from the boundaries”
Line 399 – “taking into consideration the presence of” to “relative to”
Line 430, 431, 589, 594 – “offset” to “aperture”
Line 453 – “were inferred” to “were evaluated”
Line 454 – “have been inferred” to “has been evaluated”
Line 461 – “overestimated” to “subject to wide variation”
Line 473 – “reveal” to “suggest”
Line 523 – “takes into account” to “considers”
Line 583 – “firstly, and most important” to “primarly”
Line 586, 1119, 1121 – “deformation” to “slip”
Line 590 – “additionally” to “also”
Line 611 – “considering the” to “for”
Line 613 – “are” to “were”
Line 621 – “to fulfill the forecasted” to “of fulfilling the forecast”
Line 622 – “during the” to “over”
Line 625 – “can be used to” to “would more likely”
Line 637 – “Assuming that” to “If”
Line 719 – “The Spearman correlation coefficient for the former ranges between 0.78 and 0.91, for the latter between 0.28 and 0.54 and for the stimulation cost between 0.02 and 0.13” to “The Spearman correlation coefficients range from 0.78 to 0.91 and 0.28 to 0.54, respectively, and 0.02 and 0.13 for the stimulation costs.”
Line 724 – “The Spearman correlation coefficient for the former ranges between 0.54 and 0.70, for the latter between 0.59 and 0.73, and for the stimulation cost between 0.11 and 0.20” to “The Spearman correlation coefficients vary between 0.54 and 0.70 and 0.59 and 0.73, respectively, and between 0.11 and 0.20 for the stimulation cost”
Line 765 – “the business-as-usual, where space heating is provided by oil furnaces and electricity by diesel power plants” to “current oil furnaces for space heating and electricity by diesel power plants for electricity.”
Line 768, 770 – “and” to “for”
Line 784 – “deeper” to “detailed”
Line 819 – “stored” to “store”
Line 820 – “for” to “to”
Line 839 – “with the business-as-usual” to “to the current use of fossil fuels”
Line 856 – “widest” to “most widely”
Line 861 – “have been supporting” to “have supported”
Line 870 – “can be foreseen” to “could”
Line 871 – “variability between” to “effects of”
Line 884 – “it is worth doing additional work to improve these predictions with further geothermal exploration” to “additional geothermal exploration to improve these predictions is warranted”
Line 916 – “Although these were not conducted in the scope of this study, future work can be envisioned to carry out laboratory experiments and modify the input parameters accordingly” to “Laboratory tests were not in the scope of this study but may be beneficial to refine the range of applicable input parameters”
Line 923 – “associated” to “attributable”
Line 953 – “neither” to “or”
Line 976 – “that is likely the case in” to “to be in line with anticipated”
Line 980 – “lower than” to “less expensive than”
Line 997 – “Nevertheless, it is convenient to highlight” to “Note”
Line 998 – “in those aforementioned studies” to “described by Tester et al. and Augustine”
Line 1016 – “these” to “the quality of these”
Line 1105 – “to improve” to “for improving”
R: Very interesting paper. The paper transmits all the work behind the paper. A very high scientific approach to a challenging issue.
Reviewer 2 Report
- The title of the manuscript should be more sharp.
- The motivation and contribution of this works must be clarified in the introduction section.
- More explanation is required in results and discussion section.
- For more contribution, the authors should compare their results with the related results in other published works.
- There are many grammatical errors and typoerros throughout the whole manuscript? The paper should be rechecked.
- The novelty of the work must be clearly addressed and discussed, compare your research with existing research findings and highlight novelty, (compare your work with existing research findings and highlight novelty).
- Conclusion: Future scope of the work should be provided clearly.
- The literature review section is very weak. As they are many recent published papers on the same topic need to be included. The authors can include the following relevant published articles:
https://doi.org/10.1016/j.energy.2019.04.094
https://doi.org/10.3390/su10124486
https://doi.org/10.1016/j.applthermaleng.2019.03.021
Author Response
Dear reviewer,
We would like to thank your contribution. Your comments were carefully examined and incorporated in the new version of the manuscript. In the following lines, we have addressed each specific comment and question, indicating how the modifications were incorporated in the manuscript.
Moderate English changes
The English language and style were revised by native-English colleagues, and we believe that the language and style have been significantly improved. The changes made are listed below:
Line 13 – “circulated” to “circulating fluids in”
Line 15 – “(…) large uncertainty still exists due to the current poor knowledge of both relevant geology and thermo-hydro-mechanical data.” to “(…) poor knowledge of relevant geology and thermo-hydro-mechanical data introduces significant uncertainty in numerical simulations.”
Line 16 – “Nevertheless” to “Here”
Lines 17 – deleted “provide a range of possibilities and (…) to (…)”.
Line 19 – “that provides” to “Each possibility meets (…)”
Line 26, 73, 840, 1107 – “bounded by” to “subject to”.
Line 32 – “clean” to “an environmentally benign”
Line 37 – “trend” to “predominance”
Line 38 – “watchwords” to “buzzwords”
Line 61 – “detailed yet broadly based prediction” to “comprehensive estimation”.
Line 63 – “in this physiographic region” to “the Canadian Shield”
Line 76 – “and neither was” to “nor were they”
Line 81, 82 – “lower than” to “below”
Line 84 – “such systems” to “engineered geothermal energy systems”
Line 276 – “The most significant aspect of this new sliding/opening law is that the rate of opening with displacement can be made to reduce (…)” to “The rate of opening with displacement associated with this new sliding/opening law is (…)”
Line 280 – “becomes large enough” to “exceeds threshold”
Line 332 – “(…) is now allowing (…) of microsesimic events of appropriate magnitude and in appropriate numbers.” to “(…) allows (…) of microsesimic events of appropriate magnitude and numbers (…)”
Line 357 – “A (…) to compensate for the flow overestimation typically given by the parallel plate assumption when applied to rough natural fractures. Experiments have suggested that the mechanical aperture of rough fracture overestimates the flow capacity to some extent.” To “Experiments have suggested that the mechanical aperture of rough fracture overestimates the flow capacity due to the parallel plate assumption. Therefore, a (…)”
Line 374 – “is cooled own” to “cools”
Line 388 – “allowed” to “are able”
Line 391 – “never gets close” to “happens away from the boundaries”
Line 399 – “taking into consideration the presence of” to “relative to”
Line 430, 431, 589, 594 – “offset” to “aperture”
Line 453 – “were inferred” to “were evaluated”
Line 454 – “have been inferred” to “has been evaluated”
Line 461 – “overestimated” to “subject to wide variation”
Line 473 – “reveal” to “suggest”
Line 523 – “takes into account” to “considers”
Line 583 – “firstly, and most important” to “primarly”
Line 586, 1119, 1121 – “deformation” to “slip”
Line 590 – “additionally” to “also”
Line 611 – “considering the” to “for”
Line 613 – “are” to “were”
Line 621 – “to fulfill the forecasted” to “of fulfilling the forecast”
Line 622 – “during the” to “over”
Line 625 – “can be used to” to “would more likely”
Line 637 – “Assuming that” to “If”
Line 719 – “The Spearman correlation coefficient for the former ranges between 0.78 and 0.91, for the latter between 0.28 and 0.54 and for the stimulation cost between 0.02 and 0.13” to “The Spearman correlation coefficients range from 0.78 to 0.91 and 0.28 to 0.54, respectively, and 0.02 and 0.13 for the stimulation costs.”
Line 724 – “The Spearman correlation coefficient for the former ranges between 0.54 and 0.70, for the latter between 0.59 and 0.73, and for the stimulation cost between 0.11 and 0.20” to “The Spearman correlation coefficients vary between 0.54 and 0.70 and 0.59 and 0.73, respectively, and between 0.11 and 0.20 for the stimulation cost”
Line 765 – “the business-as-usual, where space heating is provided by oil furnaces and electricity by diesel power plants” to “current oil furnaces for space heating and electricity by diesel power plants for electricity.”
Line 768, 770 – “and” to “for”
Line 784 – “deeper” to “detailed”
Line 819 – “stored” to “store”
Line 820 – “for” to “to”
Line 839 – “with the business-as-usual” to “to the current use of fossil fuels”
Line 856 – “widest” to “most widely”
Line 861 – “have been supporting” to “have supported”
Line 870 – “can be foreseen” to “could”
Line 871 – “variability between” to “effects of”
Line 884 – “it is worth doing additional work to improve these predictions with further geothermal exploration” to “additional geothermal exploration to improve these predictions is warranted”
Line 916 – “Although these were not conducted in the scope of this study, future work can be envisioned to carry out laboratory experiments and modify the input parameters accordingly” to “Laboratory tests were not in the scope of this study but may be beneficial to refine the range of applicable input parameters”
Line 923 – “associated” to “attributable”
Line 953 – “neither” to “or”
Line 976 – “that is likely the case in” to “to be in line with anticipated”
Line 980 – “lower than” to “less expensive than”
Line 997 – “Nevertheless, it is convenient to highlight” to “Note”
Line 998 – “in those aforementioned studies” to “described by Tester et al. and Augustine”
Line 1016 – “these” to “the quality of these”
Line 1105 – “to improve” to “for improving”
- The title of the manuscript should be more sharp.
The title has been modified following the reviewer comment. The new title is: “Are engineered geothermal energy systems a viable solution for arctic off-grid communities? A techno-economic study”. We believe that this new title is now sharper.
- The motivation and contribution of this works must be clarified in the introduction section.
This study was motivated by the lack of clean energy supply in the majority of the Canadian remote northern communities. The goal was to assess if deep geothermal energy harvested by engineered geothermal energy systems, or Enhanced Geothermal Systems, is a technical and economical viable alternative solution to offset the diesel consumption in such communities. The study was undertaken in the off-grid settlement of Kuujjuaq (Nunavik, Canada) to provide an example for the remaining off-grid communities. The study here described, and the results obtained, although highly speculative due to the lack of deep geothermal exploratory boreholes in the study area, represent an important contribution to understand the potential that deep geothermal energy sources have to offer in the energy transition of diesel-based regions. Furthermore, this study aims at predicting the performance of Enhanced Geothermal Systems in a location that has great geothermal data gaps. This may raise awareness about the potential of geothermal energy in areas considered at first sight unviable and trigger the interest for further geothermal developments. In fact, it is highlighted throughout the manuscript the uncertain parameters that need further gathering of information as well as possible operational strategies to develop engineered geothermal energy systems in the communities settled in the old Canadian Shield.
This paragraph was added to the end of the Introduction section (lines 84-100).
- More explanation is required in results and discussion section.
The results of this work can be divided into three main parts. The first aims at studying the initial (pre-stimulation) fracture aperture, the fractures shear stiffness, the induce shear displacement after stimulation and which fracture sets will shear and at which stimulation pressure. Then, simulations were carried out to assess operational strategies to develop EGS in Kuujjuaq according to the geological and thermo-hydro-mechanical uncertainties. Finally, the levelized cost of energy was evaluated to study if a EGS can be cost competitive compared to diesel. It should be noted that the best- and worst-case scenarios used for the EGS numerical simulations were defined following a deterministic approach by gathering all the best and worst values. However, the authors are aware that having, for example, all the best case values coming together at the same time to form a valid best-case scenario are vanishingly small. Nevertheless, although the occurrence of the best- and worst-case scenarios in nature be relatively of low probability, these scenarios correspond to the extreme cases. This is now mentioned in lines 203-205 and 398-400. Although the first part of the results seems rather trivial, this is important to study not only the subsurface characteristics pre- and post-stimulation (i.e., initial fracture aperture, fracture shear stiffness, fracture shear displacement and slipped fractures) but also to assess how the geological and thermo-hydro-mechanical subsurface uncertainty will influence the operational strategies to develop EGS and how it impacts the performance of such technologies. An important aspect highlighted by the numerical simulations is the little role played by the fault itself in the fluid flow circulation – the results obtained are relatively insensitive to the assumed initial fault aperture. If the fault segment runs quasi-parallel to the maximum principal stress (NE-SW) could be capable of further stimulations and, thus, make a huge difference in the simulation results. This is now mentioned in lines 1055-1063. Another important aspect highlighted was the role of the subsurface temperature. Greater temperature values will possibly allow the cogeneration of heat and power. This not only increases the sustainability and efficiency of an EGS in remote northern communities but has a significant economic as well. Additionally, the discussion section was extended to include a comparison of the results with other deep geothermal energy projects and with other renewable energy technologies (lines 952-1037)
- For more contribution, the authors should compare their results with the related results in other published works.
Engineered geothermal energy systems, or Enhanced Geothermal Systems (EGS), are always site-specific both in terms of the sub-surface environment and in terms of the local energy demand and price sensitivity and, thus, the comparison of the results obtained in this study with other deep geothermal energy projects developed in different geologic – economic contexts is challenging. The thermo-hydro-mechanical characteristics of the subsurface, which highly influence the design and development of EGS, vary from place to place. To date, the techno and economic feasibility of EGS in the Canadian Shield has not yet been fully assessed, and lack of field experiments removes the possibility of history matching to calibrate the numerical simulations and restrict the range of assumptions made in this study. Nevertheless, in order to design an EGS for Kuujjuaq, the approach was strongly influenced by the results obtained and the lessons learned from the hydraulically stimulated geothermal projects of Fenton Hill, Rosemanowes, Soultz and Rittershoffen (e.g., Brown et al., 2012; Parker, 1999; Richards et al., 1994; Genter et al., 2010) many of which are incorporated into the design of the FRACSIM3D model itself. For example, the acceptable targets chosen for water loss, thermal impedance and thermal drawdown were based on those adopted for those geothermal projects (e.g., Brown et al., 2012; Parker, 1999; Richards et al., 1994; Genter et al., 2010) even though the economics at Kuujjuaq are quite different. The target flow rates applied to the numerical models took also into account the values targeted and observed in the commercial projects of Soultz and Rittershoffen (Mouchot et al., 2018). The effective stimulation pressure required to induce shear slip in the pre-existing natural fractures is also found to be comparable to the values found in those four geothermal sites (Xie et al., 2015). The potential heat and power output from an EGS developed in Kuujjuaq is additionally similar to the power generated and heat produced at Soultz and Rittershoffen (Mouchot et al., 2018). Finally, the capital cost and levelized cost of energy are within values published for northern Canada and southern Quebec (Majorowicz and Grasby, 2014; Richard, 2016).
This paragraph was added to a new subsection (6.1. Comparison with other deep geothermal energy projects – lines 952-996) within the discussion section.
- There are many grammatical errors and typoerros throughout the whole manuscript? The paper should be rechecked.
The text was checked by native-English colleagues, and we believe that the language and style have improved.
- The novelty of the work must be clearly addressed and discussed, compare your research with existing research findings and highlight novelty, (compare your work with existing research findings and highlight novelty).
To date, the techno and economic feasibility of EGS in the Canadian Shield has not yet been fully assessed. The study presented in this manuscript is, thus, itself a novelty. The Canadian Shield has been considered with low geothermal potential due to the low heat flow (e.g., Jessop, 2014). However, as indicated by, for instance, Grasby et al. (2012), the Canadian Shield may be a potential target for EGS. Therefore, a techno-economic study, even if speculative and uncertain, is needed to understand if EGS is technically feasible and cost-competitive for the off-grid communities heavily relying on diesel. This is mentioned in the Introduction section (lines 45-54), in the discussion section (lines 760-770) and in the conclusions section (lines 1039-1049). Additionally, a comparison between the results obtained in the manuscript with the results from other deep geothermal energy projects was added to the discussion section (lines 952-996). However, it is important to highlight that EGS is always site-specific both in terms of the subsurface environment and in terms of the local energy demand and price sensitivity.
- Conclusion: Future scope of the work should be provided clearly.
We believe that the study carried out in this manuscript identifies questions that need to be resolved by future work. The possibilities presented, we believe, are sufficient to justify further geothermal exploration in the remote northern communities. A paragraph describing the future scope was added at the end of the conclusions section (lines 1087-1097) indicating the next steps to take. Exploratory boreholes deeper than 1 km are required for more precise heat flux and subsurface temperature assessments (e.g., Brown et al., 2012; Parker, 1999; Richards et al., 1994; Genter et al., 2010; Reinecker et al., 2021; Somma et al., 2021). These boreholes are also needed to carry out hydraulic tests (e.g., Brown et al., 2012; Parker, 1999; Richards et al., 1994; Genter et al., 2010; Reinecker et al., 2021) and stress measurements (e.g., Brown et al., 2012; Parker, 1999; Richards et al., 1994; Genter et al., 2010; Reinecker et al., 2021). Borehole televiewer can also be useful to have a better characterization of the underground fracture networks (e.g., Brown et al., 2012; Parker, 1999; Richards et al., 1994; Genter et al., 2010; Reinecker et al., 2021). The effect of temperature (or thermal lift) on the hydraulic properties (e.g., Operacz et al., 2020) can be additionally evaluated. Thus, the next step that is justified by research is the drilling of exploration boreholes to obtain more accurate data and history matching for the numerical simulations.
- The literature review section is very weak. As they are many recent published papers on the same topic need to be included. The authors can include the following relevant published articles:
https://doi.org/10.1016/j.energy.2019.04.094
https://doi.org/10.3390/su10124486
https://doi.org/10.1016/j.applthermaleng.2019.03.021
A literature review subsection (2.1. Engineered/enhanced geothermal systems) was added to a new background information section (2. Background information) reviewing the sites where hydraulic stimulation treatments have been applied and the lessons learned that helped to design the EGS presented in this study. This review is in lines 102-134. Additionally, a review of the research carried out in Nunavik in general (2.2. Nunavik’s geothermal potential) and in Kuujjuaq in particular (3.2. Previous research undertaken in Kuujjuaq) can be found in lines 135-154 and lines 186-228, respectively.
Additionally, a new subsection was added to the discussion (6.2. Comparison with other renewable energy alternative) mentioning that beyond EGS, ground-coupled heat pumps and borehole thermal energy storage can be another geothermal solution to reduce diesel consumption in remote northern regions. The relevant published articles referred by the reviewer were cited in lines 1000-1005. However, it is important to highlight that the extracted energy with such shallow systems (100 to 300 m) is generally insufficient to fulfill the heating demand of a single residential dwelling, relying on an energy input to cover the remaining load. This new subsection also reviews other possible renewable energy technologies that have been proposed for northern communities. These solutions include biomass, wind, solar, biofuels and hydro. The pros and cons of each of these technologies are described in lines 1010-1037.
References
Jessop, A.M. Geothermal Energy. Available online: https://www.thecanadianencyclopedia.ca/en/article/geothermal-energy
Grasby, S.E.; Allen, D.M.; Bell, S.; Chen, Z.; Ferguson, G.; Jessop, A.; Kelman, M.; Ko, M.; Majorowicz, J.; Moore, M.; Raymond, J., Therrien, R. Geothermal Energy Resource Potential of Canada, Report No.: Open File 6914; Geological Survey of Canada: Ottawa, ON, Canada, 2012; 322 p.
Brown, D.W.; Duchane, D.V.; Heiken, G.; Hriscu, V.T. Mining the Earth’s Heat: Hot Dry Rock Geothermal Energy; Springer: Berlin, Germany; 669 p.
Parker, R. The Rosemanowes HDR project 1983-1991. Geothermics 1999, 28(4-5), 603-615.
Richards, H.G.; Parker, R.H.; Green, A.S.P.; Jones, R.H.; Nicholls, J.D.M.; Nicol, D.A.C.; Randall, M.M.; Richards, S.; Stewart, R.C.; Willis-Richards, J. The performance and characteristics of the experimental hot dry rock geothermal reservoir at Rosemanowes, Cornwall (1985-1988). Geothermics 1994, 23(2), 73-109.
Genter, A.; Evans, K.; Cuenot, N.; Fritsch, D.; Sanjuan, B. Contribution of the exploration of deep crystalline fractured reservoir of Soultz to the knowledge of enhanced geothermal systems (EGS). Comptes Rendus Geoscience 2010, 342(7-8), 502-516.
Mouchot, J.; Genter, A.; Cuenot, N.; Scheiber, J.; Seibel, O.; Bosia, C.; Ravier, G. First year of operation from EGS geothermal plants in Alsace, France: scaling issues. Proceedings of the 43rd Workshop on Geothermal Reservoir Engineering, Stanford, CA, US, February 12-14, 2018.
Xie, L.; Min, K.-B.; Song, Y. Observations of hydraulic stimulations in seven enhanced geothermal system projects. Renewable Energy 2015, 79, 56-65.
Reinecker, J.; Gutmanis, J.; Foxford, A.; Cotton, L.; Dalby, C.; Law, R. Geothermal exploration and reservoir modelling of the United Downs deep geothermal project, Cornwall (UK). Geothermics 2021, 97, 102226.
Somma, R.; Blessent, D.; Raymond, J.; Constance, M.; Cotton, L.; Natale, G.; Fedele, A.; Jurado, M.J.; Marcia, K.; Miranda, M.M.; Troise, C.; Wiersberg, T. Review of recent drilling projects in unconventional geothermal resources at Campi Flegrei Caldera, Cornubian Batholith, and Williston Sedimentary Basin. Energies 2021, 14(11), 3306.
Operacz, A.; Bielec, B.; Tomaszewska, B.; Kaczmarczyk, M. Physicochemical composition variability and hydraulic conditions in a geothermal borehole – The latest study in Podhale Basin, Poland. Energies 2020, 13(15), 3882.
Majorowicz, J.; Grasby, S.E. Geothermal energy for northern Canada: is it economical? Natural Resources Research 2014, 23(1), 159-173.
Richard, M.-A. Production d’électricité avec des systèmes géothermiques stimulés au Québec : analyse des résultats d’un outil de simulation, Report No.: IREQ-2016-0001; Hydro Québec Institut de recherche: Varennes, QC, Canada, 2016; 164 p.
Reviewer 3 Report
The paper brings novelty, seems to be a good example of scientific article. The main course of calculation is described widely and properly. Authors have made an excellent work. In my opinion there is only a lack of comparison theoretical results based on modelling with real results from existing boreholes. The verification of results obtained from calculation with real parameters is in my opinion necessary. The Introduction chapter should be expanded with characteristic of deep geothermal boreholes in the study area or in other areas in the world. Specific comments are in the pdf files. Summarized the paper needs only a minor revision and could be published.

Author Response
Dear reviewer,
We would like to thank your contribution. Your comments were carefully examined and incorporated in the new version of the manuscript. In the following lines, we have addressed each specific comment and question, indicating how the modifications were incorporated in the manuscript.
The paper brings novelty, seems to be a good example of scientific article. The main course of calculation is described widely and properly. Authors have made an excellent work. In my opinion there is only a lack of comparison theoretical results based on modelling with real results from existing boreholes. The verification of results obtained from calculation with real parameters is in my opinion necessary.
Engineered geothermal energy systems, or Enhanced Geothermal Systems (EGS), are always site-specific both in terms of the sub-surface environment and in terms of the local energy demand and price sensitivity and, thus, the comparison of the results obtained in this study with other deep geothermal energy projects developed in different geologic – economic contexts is challenging. The thermo-hydro-mechanical characteristics of the subsurface, which highly influence the design and development of EGS, vary from place to place. To date, the techno and economic feasibility of EGS in the Canadian Shield has not yet been fully assessed, and lack of field experiments removes the possibility of history matching to calibrate the numerical simulations and restrict the range of assumptions made in this study. Nevertheless, in order to design an EGS for Kuujjuaq, the approach was strongly influenced by the results obtained and the lessons learned from the hydraulically stimulated geothermal projects of Fenton Hill, Rosemanowes, Soultz and Rittershoffen (e.g., Brown et al., 2012; Parker, 1999; Richards et al., 1994; Genter et al., 2010) many of which are incorporated into the design of the FRACSIM3D model itself. For example, the acceptable targets chosen for water loss, thermal impedance and thermal drawdown were based on those adopted for those geothermal projects (e.g., Brown et al., 2012; Parker, 1999; Richards et al., 1994; Genter et al., 2010) even though the economics at Kuujjuaq are quite different. The target flow rates applied to the numerical models took also into account the values targeted and observed in the commercial projects of Soultz and Rittershoffen (Mouchot et al., 2018). The effective stimulation pressure required to induce shear slip in the pre-existing natural fractures is also found to be comparable to the values found in those four geothermal sites (Xie et al., 2015). The potential heat and power output from an EGS developed in Kuujjuaq is additionally similar to the power generated and heat produced at Soultz and Rittershoffen (Mouchot et al., 2018). Finally, the capital cost and levelized cost of energy are within values published for northern Canada and southern Quebec (Majorowicz and Grasby, 2014; Richard, 2016).
This paragraph was added to a new subsection (6.1. Comparison with other deep geothermal energy projects – lines 952-996) within the discussion section.
The Introduction chapter should be expanded with characteristic of deep geothermal boreholes in the study area or in other areas in the world.
Nunavik’s geothermal energy potential has been investigated by Majorowicz and Minea (2015) and Comeau et al. (2017). These regional-scale studies are based on scarce and sparse data distribution as can be seen in the new Figure 1. In Nunavik, a territory of 507 000 km2, has only 3 deep boreholes with heat flow assessment and all of these lie at distance of 430 to 500 km from Kuujjuaq. The heat flow from these boreholes is in the range 22 to 38 mW m-2. This information is now provided in the new subsection (2.2. Nunavik’s geothermal potential – lines 135-154). Additionally, in subsection 3.2. Previous research undertaken in Kuujjuaq (lines 186-228) is explained with all the research carried out in Kuujjuaq, at a local-scale, that constitute the basis for the numerical simulations carried out in the present manuscript.
Specific comments are in the pdf files. Summarized the paper needs only a minor revision and could be published.
- It seems to be too long
The title was shortened, and we believe is now sharper and straight to the point. The new title is: “Are engineered geothermal energy systems a viable solution for arctic off-grid communities? A techno-economic study”.
- deep geothermal potential doesn't follow the meteorologic parameters. This chapter must be expand for characteristic of hydrogeological details. Profiles, cross-sections, geothermal gradient/step and others which could bring more specific information about geothermal reservoir. Is there any geothermal boreholes or similar instalation? Please show them on a map and desribe details of their works
The meteorologic parameters are described to provide background information about the harsh climate feel in the community that justifies the high annual heating energy demand of each residential dwelling. This information is important to evaluate the amount of energy that needs to be extracted with an EGS.
A new subsection (2.2. Nunavik’s geothermal potential – lines 135-154) illustrates the location of deep boreholes in the province of Quebec and provides background information about the heat flow assessed for the region. As the new Figure 1 shows, the data available north of the 55º parallel is scarce and sparse. Additionally, in subsection 3.2. Previous research undertaken in Kuujjuaq (lines 186-228) is described first-order estimates of heat flux, subsurface temperature, hydraulic and stress regime in the community.
- The theoretical calculation are proper but in my opinion there is a little lack of comparison with working installations. For example the temperature effect is not under consideration but it could be important in the deep boreholes.
There is no working installation to date in Kuujjuaq and no study has been carried out to date evaluating the techno-economic potential of EGS in the Canadian Shield. This lack of data greatly limits the history matching and calibration of assumptions and results. Nevertheless, a new subsection (6.1. Comparison with other deep geothermal energy projects – lines 952-996) was added to the discussion section comparing the results of this study with the results from other deep geothermal energy projects.
The temperature effect (or thermal lift) was not considered and is beyond the scope of this work but is now mentioned in the future scope paragraph at the end of the conclusions (lines 1094-1095).
- In my opinion in Discussion chapter there is a lack of comparison with working deep geothermal energy plants. If there is no realized projects with geothermal energy, Authors should compare results from modelling with chosen projects from the world. As an example the Podhale basin in south Poland is a area with several deep geothermal boreholes. There are many published papers with results/environmental impact and similar: https://doi.org/10.3390/en13102495; https://doi.org/10.3390/en13153882; DOI: 10.24425/aep.2019.127985; DOI: DOI10.7494/geol.2018.44.4.379
A comparison section was added to the discussion (lines 952-996) where our results are compared with other deep geothermal energy projects, namely with the research undertaken in the Podhale basin in Poland. In fact, the thermodynamic calculations carried out by Kaczmarczyk et al. (2020) using the C-PIG-1 well (Malopolskie Voivodship, southern Poland) which has a temperature (82 ºC) and flow rate (51.22 kg s-1) suggest that, if the same order of values are found in Kuujjuaq (i.e., gross electricity generation not exceeding 1.9 GWh/year for the Organic Rankine Cycle and 3.5 GWh/year for the Kalina Cycle), then electricity generation from geothermal water of 82 ºC and flow rates of 51 kg s-1 does not seem a viable option to replace diesel since the annual average electricity demand in the community is almost 19 GWh.
- need extensions for papers about real working geothermal boreholes to compare results from calculation with measured parameters in the field.
The references were updated with papers describing real working geothermal boreholes. For example, the Fenton Hill venture is described in great detail by Brown et al. (2012) and a summary of the lessons learned can be read in Kelkar et al. (2016). A compilation of the development phases of the Rosemanowes geothermal project, problems faced, and unresolved issues are provided by, for example, Parker (1999). Richards et al. (1994) discuss the performance and characteristics of the Rosemanowes hydraulically stimulated geothermal reservoir. The authors also discuss the fundamental parameters controlling the impedance, thermal performance and water losses. The contribution of the Soultz project for the scientific community and its development phases are described in detail by, for instance, Genter et al. (2010). The lessons learned from past geothermal projects employing hydraulic stimulation treatments to crystalline rocks were taken into consideration in this study.
Moreover, the future scope of this work (lines 1087-1097) will rely on lessons learned from geothermal boreholes. Exploratory boreholes deeper than 1 km are required for accurate heat flux and subsurface temperature assessments (e.g., Brown et al., 2012; Parker, 1999; Richards et al., 1994; Genter et al., 2010; Reinecker et al., 2021; Somma et al., 2021) These boreholes are also needed to carry out hydraulic tests (e.g., Brown et al., 2012; Parker, 1999; Richards et al., 1994; Genter et al., 2010; Reinecker et al., 2021) and stress measurements (e.g., Brown et al., 2012; Parker, 1999; Richards et al., 1994; Genter et al., 2010; Reinecker et al., 2021). Borehole televiewer can also be useful to have a better characterization of the underground fracture networks (e.g., Brown et al., 2012; Parker, 1999; Richards et al., 1994; Genter et al., 2010; Reinecker et al., 2021). The effect of temperature (or thermal lift) on the hydraulic properties (e.g., Operacz et al., 2020) can be additionally evaluated.
References
Brown, D.W.; Duchane, D.V.; Heiken, G.; Hriscu, V.T. Mining the Earth’s Heat: Hot Dry Rock Geothermal Energy; Springer: Berlin, Germany; 669 p.
Parker, R. The Rosemanowes HDR project 1983-1991. Geothermics 1999, 28(4-5), 603-615.
Richards, H.G.; Parker, R.H.; Green, A.S.P.; Jones, R.H.; Nicholls, J.D.M.; Nicol, D.A.C.; Randall, M.M.; Richards, S.; Stewart, R.C.; Willis-Richards, J. The performance and characteristics of the experimental hot dry rock geothermal reservoir at Rosemanowes, Cornwall (1985-1988). Geothermics 1994, 23(2), 73-109.
Genter, A.; Evans, K.; Cuenot, N.; Fritsch, D.; Sanjuan, B. Contribution of the exploration of deep crystalline fractured reservoir of Soultz to the knowledge of enhanced geothermal systems (EGS). Comptes Rendus Geoscience 2010, 342(7-8), 502-516.
Mouchot, J.; Genter, A.; Cuenot, N.; Scheiber, J.; Seibel, O.; Bosia, C.; Ravier, G. First year of operation from EGS geothermal plants in Alsace, France: scaling issues. Proceedings of the 43rd Workshop on Geothermal Reservoir Engineering, Stanford, CA, US, February 12-14, 2018.
Xie, L.; Min, K.-B.; Song, Y. Observations of hydraulic stimulations in seven enhanced geothermal system projects. Renewable Energy 2015, 79, 56-65.
Kaczmarczyk, M.; Tomaszewska, B.; Operacz, A. Sustainable utilization of low entalphy geothermal resources to electricity generation through a cascade system. Energies 2020, 13(10), 2495.
Kelkar, S.; Gabriel, G.W.; Rehfeldt, K. Lessons learned from the pioneering hot dry rock project at Fenton Hill, USA. Geothermics 2016, 63, 5-14.
Reinecker, J.; Gutmanis, J.; Foxford, A.; Cotton, L.; Dalby, C.; Law, R. Geothermal exploration and reservoir modelling of the United Downs deep geothermal project, Cornwall (UK). Geothermics 2021, 97, 102226.
Somma, R.; Blessent, D.; Raymond, J.; Constance, M.; Cotton, L.; Natale, G.; Fedele, A.; Jurado, M.J.; Marcia, K.; Miranda, M.M.; Troise, C.; Wiersberg, T. Review of recent drilling projects in unconventional geothermal resources at Campi Flegrei Caldera, Cornubian Batholith, and Williston Sedimentary Basin. Energies 2021, 14(11), 3306.
Operacz, A.; Bielec, B.; Tomaszewska, B.; Kaczmarczyk, M. Physicochemical composition variability and hydraulic conditions in a geothermal borehole – The latest study in Podhale Basin, Poland. Energies 2020, 13(15), 3882.
Round 2
Reviewer 2 Report
This work could be accepted for publication with present modified version.